# SFT Doesn't Always Hurt General Capabilities: Revisiting Domain-Specific Fine-Tuning in LLMs

**Jiacheng Lin**[1,†]    **Zhongruo Wang**[2,†]    **Kun Qian**[2]    **Tian Wang**[2]    **Arvind Srinivasan**[2]

**Hansi Zeng**[3]        **Ruochen Jiao**[2]        **Xie Zhou**[2]        **Jiri Gesi**[2]        **Dakuo Wang**[6]

**Yufan Guo**[2]        **Kai Zhong**[2]        **Weiqi Zhang**[2]        **Sujay Sanghavi**[4]    **Changyou Chen**[2,5]

**Hyokun Yun**[2]        **Lihong Li**[2]

[1]University of Illinois Urbana-Champaign    [2]Amazon    [3]University of Massachusetts Amherst
[4]University of Texas at Austin    [5]University at Buffalo    [6]Northeastern University

## Abstract

Supervised Fine-Tuning (SFT) on domain-specific datasets is a common approach to adapt Large Language Models (LLMs) to specialized tasks but is often believed to degrade their general capabilities. In this work, we revisit this trade-off and present both empirical and theoretical insights. First, we show that SFT does not always hurt: using a smaller learning rate can substantially mitigate general performance degradation while preserving comparable target-domain performance. We then provide a theoretical analysis that explains these phenomena and further motivates a new method, *Token-Adaptive Loss Reweighting* (TALR). Building on this, and recognizing that smaller learning rates alone do not fully eliminate general-performance degradation in all cases, we evaluate a range of strategies for reducing general capability loss, including L2 regularization, LoRA, model averaging, FLOW, and our proposed TALR. Experimental results demonstrate that while no method completely eliminates the trade-off, TALR consistently outperforms these baselines in balancing domain-specific gains and general capabilities. Finally, we distill our findings into practical guidelines for adapting LLMs to new domains: (i) using a small learning rate to achieve a favorable trade-off, and (ii) when a stronger balance is further desired, adopt TALR as an effective strategy.

## 1 Introduction

Large Language Models (LLMs) have demonstrated remarkable performance across a wide range of general-purpose tasks, including question answering, mathematical reasoning, and code generation (Yang et al., 2024; 2025; Touvron et al., 2023; Dubey et al., 2024). To further enhance their effectiveness in specialized applications, practitioners often perform additional supervised fine-tuning (SFT) using domain-specific data. This process enriches the model with domain knowledge and yields substantial performance gains on domain-specific tasks (Labrak et al., 2024; Lin et al., 2024; Peng et al., 2024). SFT has thus become a standard paradigm for adapting LLMs to real-world deployment scenarios.

However, recent studies have shown that fine-tuning LLMs on domain-specific datasets can substantially impair their generalization capabilities (Huan et al., 2025; Lin et al., 2025a; Chen et al., 2025; Bansal & Sanghavi, 2025; Sanyal et al., 2025; Chu et al., 2025; Shenfeld et al., 2025). For example,

---

[†]Equal contributions. Corresponding to `jl254@illinois.edu`. This work was done during Jiacheng Lin and Hansi Zeng's internship at Amazon.

performing SFT on LLMs like Qwen-3 (Yang et al., 2025) or Gemma-3 (Team et al., 2025) using domain-specific datasets, such as those from e-commercial or biomedical domains, often leads to significant performance drops on general-purpose benchmarks such as GSM8K (Cobbe et al., 2021), HumanEval (Chen et al., 2021), or IFEval (Zhou et al., 2023), which assess core capabilities like mathematical reasoning, code generation, and instruction following. This phenomenon raises the need for a closer examination of domain-specific SFT.

In this work, we revisit the phenomenon of general capability degradation induced by domain-specific SFT. **Surprisingly, domain-specific SFT does not always significantly degrade general capabilities, contrary to prior claims.** Our experiments reveal that, in most cases:

> **Using a smaller learning rate allows domain-specific SFT to achieve a favorable trade-off:**
> - General-purpose performance degradation is largely mitigated;
> - Target domain performance is comparable to that with larger learning rates.

The first observation is relatively expected, since smaller learning rates naturally suppress parameter drift compared to more aggressive updates (Pareja et al., 2025). The second, however, is more surprising. Prior to the LLM era, practical experience in machine and deep learning suggested that larger learning rates are often essential for better downstream performance (Mohtashami et al., 2023; Li et al., 2019; Sadrtdinov et al., 2024). In contrast, we show that LLMs behave differently: comparable domain-specific performance can still be achieved under smaller learning rates. We further provide a theoretical analysis supporting this phenomenon. In addition, a closer inspection of prior studies shows that their strong degradation claims predominantly arise under relatively large learning rates. Taken together, our empirical and theoretical evidence demonstrates that a careful choice of learning rate offers a practical path to balance domain adaptation with general capability preservation (§3).

While adopting smaller learning rates typically yields a better trade-off, we also observe that this does not fully mitigate the general-performance degradation in all cases. To address this, we further investigate mitigation approaches that could mitigate such degradation (§4). Specifically, we assess a range of representative strategies evaluated in Sanyal et al. (2025), including L2 regularization, LoRA (Hu et al., 2022), model averaging (Wortsman et al., 2022), and FLOW (Sanyal et al., 2025), along with our proposed method, *Token-Adaptive Loss Reweighting (TALR)*. TALR adaptively down-weights hard tokens by solving a constrained optimization problem that admits a closed-form solution, thereby tempering their potential disproportionate influence on general capability degradation during training. Our experiments demonstrate that TALR provides advantages in further suppressing general-performance degradation compared to these baselines. Nevertheless, no existing method including TALR can completely eliminate the degradation, highlighting the need for more advanced strategies to be explored in future work.

We further conduct a token-level analysis to better understand domain-specific SFT (§4.3). This analysis yields two key findings: (1) **Most tokens in SFT training data pose low learning difficulty to the LLMs, even when the overall domain-specific task performance is poor.** The relatively fewer hard tokens (i.e., low-probability tokens) typically arise either from a lack of domain knowledge in the pretrained model or from stylistic mismatches between the domain-specific data and the pretrained model. (2) **TALR induces a token-level curriculum-like learning dynamic.** In the early stages of training, easier tokens receive more focus, while hard tokens are down-weighted. As training progresses, however, some of these hard tokens become relatively easier for the model, and their weights gradually increase. This dynamic allows TALR to smoothly shift focus over time, balancing the injection of domain knowledge with the preservation of general capabilities.

Finally, we summarize our findings into a practical guideline for domain-specific SFT:

> **Guidelines for domain-specific SFT.**
> - **Use a smaller learning rate** to achieve a favorable trade-off between domain performance and general-purpose capability preservation.
> - **When a stronger balance is further required**, adopt TALR as an effective strategy to further suppress general-performance degradation.

## 2   RELATED WORK

Our problem setting can be broadly framed within the scope of continual learning, where models must acquire new knowledge while retaining previously learned capabilities to avoid catastrophic forgetting. Existing approaches are typically divided into two categories: data-dependent and data-oblivious. Data-dependent methods assume access to a subset of the training data from earlier stages, whereas data-oblivious methods rely solely on the pre-trained model without revisiting any prior data. The latter is particularly realistic in practice, as access to proprietary or large-scale pre-training corpora is often infeasible, yet it remains relatively underexplored. For a broader overview of this landscape, we refer readers to recent surveys (Wang et al., 2024a). Our focus in this paper is on the **data-oblivious setting**.

**Data-oblivious approaches.** One line of work introduces loss regularization to constrain the fine-tuned model from drifting too far from its initialization, such as L2 regularization in parameter space (Kumar et al., 2025; Kirkpatrick et al., 2017). Another line of work explores the idea of model averaging, which combines the parameters of the pre-trained model and the fully fine-tuned model through a convex combination, aiming to balance adaptation with retention (Wortsman et al., 2022; Lubana et al., 2022; Ilharco et al., 2023; Kleiman et al., 2025). LoRA (Hu et al., 2022; Biderman et al., 2024) represents another widely used strategy, enforcing low-rank updates to the weight matrices so that parameter changes are confined to a restricted subspace, thereby limiting catastrophic drift while improving efficiency. Besides, data reweighting has been explored as a promising strategy; for example, FLOW (Sanyal et al., 2025) mitigates forgetting in vision tasks by adjusting the loss weights of easy and hard samples.

**Extensions to LLMs.** Existing research has primarily focused on data-dependent methods in the LLM context (Scialom et al., 2022; Yin et al., 2023; Wang et al., 2024b; Xiong et al., 2023; Mok et al., 2023). In contrast, data-oblivious approaches remain relatively underexplored, though several studies have begun adapting ideas from continual learning on traditional models to LLMs (Sanyal et al., 2025; Razdaibiedina et al., 2023; Wang et al., 2023; Zhao et al., 2024). Refer to the survey by Wu et al. (2024) for more details. However, LLMs differ substantially from earlier architectures in scale, pre-training regimes, and emergent capabilities, which makes their adaptation dynamics distinct. As a result, we revisit continual SFT of LLMs on domain-specific datasets, aiming to better understand its mechanism.

## 3   LEARNING RATE MATTERS: REVISITING ITS ROLE IN GENERAL CAPABILITY DEGRADATION DURING DOMAIN-SPECIFIC SFT

In this section, we revisit the role of learning rate in domain-specific SFT and its impact on general capability degradation. Surprisingly, we find that using a smaller learning rate (e.g., $1e{-}6$) can substantially reduce the loss of general capabilities, while achieving domain-specific task performance on par with much larger learning rates. This suggests that the severe degradation reported in prior work may stem, at least in part, from overly aggressive optimization (Huan et al., 2025; Lin et al., 2025a; Chen et al., 2025; Bansal & Sanghavi, 2025; Sanyal et al., 2025; Shenfeld et al., 2025). Indeed, many of these studies used relatively large learning rates such as $5e{-}6$ or $2e{-}5$.

To systematically investigate this effect, we experiment on two domain-specific datasets: MedCalc (Khandekar et al., 2024) and ESCI (Reddy et al., 2022). We choose these datasets because existing open-source LLMs perform poorly on them, making them representative scenarios where domain-specific SFT is most motivated: to enhance specialized capabilities in domains where the initialized model is weak. Below are details of the experimental setups and results for each dataset.

### 3.1   EXPERIMENTAL SETUP

#### 3.1.1   DATASETS

**MedCalc** (Khandekar et al., 2024) consists of 10.1k training and 1.05k test examples. Each instance includes a brief patient note and a clinical instruction (e.g., "What is the patient's $CHA_2DS_2$-VASc score?"), with the goal of predicting a numeric, categorical, or datetime answer. The training set provides gold chain-of-thought (CoT) rationales, which we use as supervision targets during SFT.

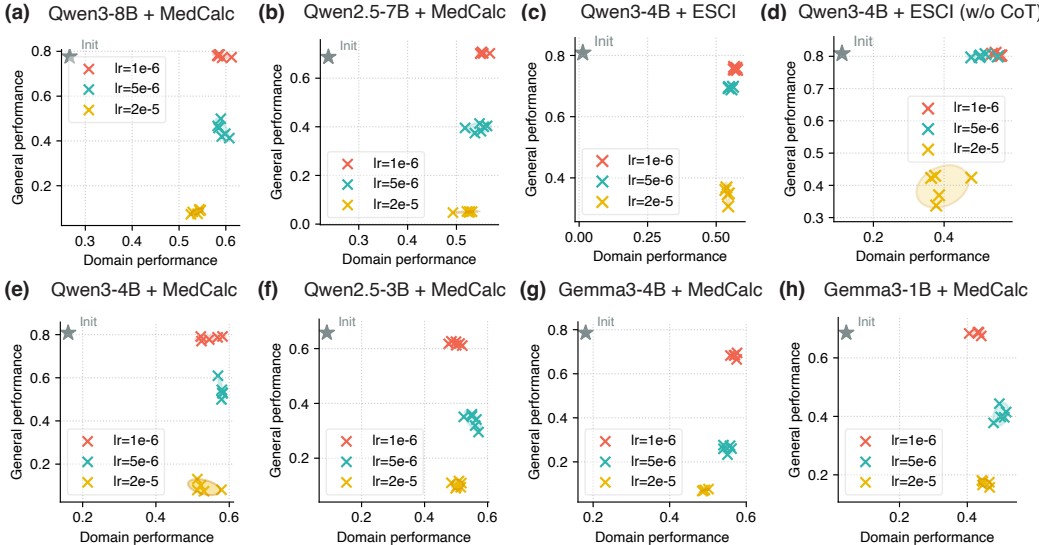

Figure 1: **Effect of learning rate on domain-specific and general capability performance during supervised fine-tuning (SFT).** We conduct experiments on two domain-specific datasets, *Med-Calc* and *ESCI*. For the *ESCI (w/o CoT)* variant, the model is trained only to predict the final label without intermediate reasoning steps, unlike the other three settings where reasoning traces are available. General capability performance is measured as the average across IFEval, GSM8K, and HumanEval unless otherwise specified. We observe that smaller learning rates yield a more favorable trade-off (upper-right corner) between domain performance and general performance.

The **ESCI** dataset (Reddy et al., 2022) is an e-commerce product classification benchmark containing query–product pairs labeled as *Exact*, *Substitute*, *Complement*, or *Irrelevant*. The training set consists of 49k examples, and the test set contains 10k examples. We consider two training settings: w/ CoT, where the target sequence includes both reasoning and the final label, and w/o CoT, where it contains only the label.

### 3.1.2 EVALUATION PROTOCOL

For the **MedCalc** task, we follow the evaluation protocol of Khandekar et al. (2024) and report accuracy based on the model's final answer. For general capability evaluation, we measure performance on a suite of general-purpose benchmarks using the lm-evaluation-harness framework (Gao et al., 2024), following the same evaluation setup as prior works (Lin et al., 2025a; Sanyal et al., 2025; Bansal & Sanghavi, 2025). Model checkpoints are selected based on their best performance on the target domain task, after which the corresponding models are evaluated on general-purpose benchmarks, reflecting practical scenarios where downstream task performance is prioritized (Sanyal et al., 2025; Bansal & Sanghavi, 2025). The evaluation metric for each benchmark is detailed in Appendix C.1. Since **ESCI** is highly imbalanced across classes, we follow prior work on imbalanced classification (Xu et al., 2024; 2025) and report *balanced accuracy* (BACC) as our primary metric.

### 3.2 MAIN RESULTS

**Finding 1: Smaller learning rates achieve a more favorable trade-off.** From Figure 1, we observe that for both *MedCalc* and *ESCI*, smaller learning rates consistently lead to points located toward the upper-right region of the plots. This indicates that they can effectively mitigate degradation in general capabilities while simultaneously delivering strong performance on the target domain tasks.

**Finding 2: Label-only supervision loosens learning rate constraints for Pareto-optimal trade-offs.** When the target sequence consists solely of the ground-truth label (e.g., <answer>[label]</answer>) without intermediate reasoning steps, the range of learning rates that achieve Pareto-optimal trade-offs becomes broader. As shown in Figure 1(d), a learn-

ing rate of $5e{-}6$ performs comparably to $1e{-}6$ in the upper-right region, which contrasts with the trend observed in the other subfigures of Figure 1.

**Remark:** From our experiments on MedCalc and ESCI, as well as the additional results and analyses in Appendix C.3, we observe consistent patterns: smaller learning rates can substantially reduce general capability degradation while maintaining competitive domain-specific performance. This naturally raises the question:

> *Why do milder updates preserve general abilities while still enabling strong domain gains?*

To shed light on this phenomenon, we next turn to a theoretical analysis, aiming to uncover insights into how the learning rate shapes the trade-off between domain adaptation and the preservation of general capabilities in domain-specific SFT.

### 3.3 Theoretical Analysis

To better understand the empirical phenomena observed previously, we provide a theoretical analysis from the perspective of information theory. Motivated by the equivalence between language modeling and data compression (Deletang et al., 2024; Ji et al., 2025), we view an LLM as a *compressor*, where the effectiveness of training can be measured through changes in code length. In this view, improvements or degradations in performance across datasets correspond to variations in compression rate. Below, we formalize this perspective by introducing the notion of token trees and describing the LLM compression protocol in our context.

**Definition 3.1** (Token Tree $\mathcal{T}$). *For a dataset $\mathcal{D} = \{z_i \in \mathcal{V}^\infty \mid i = 1, 2, \ldots\}, \quad |\mathcal{V}| < \infty$, where $\mathcal{V} = \{v_1, v_2, \ldots, v_{|\mathcal{V}|}\}$ is a finite vocabulary of size $|\mathcal{V}|$, the token tree of $\mathcal{D}$, denoted as $\mathcal{T}_\mathcal{D}$, is defined as follows: (1) each node has $|\mathcal{V}|$ child nodes labeled $v_1, v_2, \ldots, v_{|\mathcal{V}|}$, along with an end-of-sequence (EOS) leaf node; (2) The weight of a non-leaf node is the sum of the weights of all its child nodes; (3) The path from the root to an EOS leaf node defines a response $z_i$, with the corresponding EOS node weight representing the response's probability.*

**Definition 3.2** (LLM Compression Protocol). *Let $\mathcal{T}_\mathcal{D}$ be the token tree of dataset $\mathcal{D}$, and let $q_\theta(\cdot \mid u)$ denote the conditional distribution over $\mathcal{V} \cup \{\mathrm{EOS}\}$ predicted by an LLM with parameters $\theta$ at node $u \in \mathcal{T}_\mathcal{D}$. Given a response $z$ (a path from the root to an EOS leaf, truncated to a pre-defined maximum depth $d$), the LLM compression protocol encodes $z$ using arithmetic coding, where at each step the coding probabilities are given by $q_\theta(\cdot \mid u)$ for the current node $u$ along the path of $z$.*

Having established the compression protocol, we now follow prior work (Deletang et al., 2024; Ji et al., 2025) and use changes in expected code length as a surrogate metric for an LLM's modeling quality on a given dataset distribution. In this view, reductions in code length discrepancy correspond to better alignment with the data distribution. Formally, this is captured by the following proposition.

**Proposition 3.1** (Expected Code Length Discrepancy under Model Shift). *Consider two model distributions $q_{\theta_1}(\cdot)$ and $q_{\theta_2}(\cdot)$ over the token tree $\mathcal{T}_\mathcal{D}$ with distribution $P$. The change in expected code length on $P$ when shifting from $q_{\theta_1}$ to $q_{\theta_2}$ is $\Delta L(P) = \mathbb{E}_{z \sim P}[L_{q_{\theta_2}}(z)] - \mathbb{E}_{z \sim P}[L_{q_{\theta_1}}(z)] = -\sum_{l=1}^{d} \sum_j p_{l,j} \log \frac{q_{l,j}^{(2)}}{q_{l,j}^{(1)}}$. Equivalently, $\Delta L(P) = \mathrm{KL}(P \,\|\, q_{\theta_2}) - \mathrm{KL}(P \,\|\, q_{\theta_1})$.*

Based on the above, we now turn to our main goal: explaining the empirical phenomena observed in §3, namely **Finding 1** and **2**. To keep the presentation clear and concise, we provide simplified *informal* statements of our key theorems below, while the full formal versions and proofs are referred to Appendix B.

---

**Theorem 3.1.** *(Informal) Under certain assumptions, consider fine-tuning on a domain-specific dataset $\mathcal{D}_2$ with a fixed target domain improvement $\Delta_\star > 0$ (i.e., $\Delta L(P_2) \leq -\Delta_\star$). The general-performance degradation on $\mathcal{D}_1$, which is already well modeled by the LLM, admits an upper bound*

$$\Delta L(P_1) \leq k_1 \Delta_\star + k_2 \Delta_\star^2 \lambda$$

*where $\lambda$ is the effective per-step size and $k_1, k_2$ are constants determined by the model and data. Thus, using smaller steps (smaller $\lambda$) leads to strictly tighter guarantees on general-performance preservation.*

---

Table 1: Comparison of domain and general performance on the MedCalc Benchmark under learning rate 1e−6. Both **Standard SFT** (with smaller learning rate) and **TALR** are our contributions, and together they achieve the best overall trade-offs compared with the other baselines.

| Method | Qwen2.5-3B | | Qwen3-4B | | Gemma3-4B | | Average | |
|---|---|---|---|---|---|---|---|---|
| | Domain | General | Domain | General | Domain | General | Domain | General |
| **Standard (Ours)** | 0.4947 | 0.6202 | 0.5484 | 0.7837 | 0.5587 | 0.6734 | 0.5339 | 0.6924 |
| L2-Reg | 0.4904 | 0.6205 | 0.4692 | 0.7964 | 0.5595 | 0.6750 | 0.5064 | 0.6973 |
| LoRA | 0.1261 | 0.5831 | 0.1945 | 0.7640 | 0.2233 | 0.1241 | 0.1813 | 0.4904 |
| Wise-FT | 0.1948 | 0.6285 | 0.1428 | 0.7884 | 0.2573 | 0.7635 | 0.1983 | 0.7268 |
| FLOW | 0.3641 | 0.5974 | 0.4768 | 0.7870 | 0.5673 | 0.6914 | 0.4694 | 0.6920 |
| **TALR (Ours)** | 0.4806 | 0.6478 | 0.4889 | 0.7880 | 0.5338 | 0.7150 | 0.5011 | 0.7169 |

Here, $\lambda \in (0,1)$ denotes the *per-step size of the distributional update*; formal definitions are provided in Appendix B.3. In practice, a *smaller learning rate* induces a *smaller* $\lambda$. Therefore, Theorem 3.1 explains **Finding 1**: adopting a smaller learning rate (i.e., smaller $\lambda$) reduces the upper bound on general-performance degradation, consistent with the empirical trend observed in §3.

> **Theorem 3.2.** *(Informal) Under certain assumptions, fix a tolerance on general-performance degradation on $\mathcal{D}_1$ (i.e., $\Delta L(P_1) \leq \varepsilon_{\text{fg}}$). Then the maximal safe per-step size satisfies $\lambda_{\max} \propto \frac{\varepsilon_{\text{fg}}}{\sqrt{s}}$, where $s$ is the expected number of low-probability tokens per example on $\mathcal{D}_2$, defined as tokens whose probabilities under the LLM are below a threshold.*

This result explains **Finding 2**: when training with only labels, the number of hard tokens is reduced compared to training with both labels and chain-of-thought annotations, thereby increasing the safe step-size range. This explains why in our ESCI experiments, label-only SFT tolerated larger learning rates (e.g., 5e−6) without causing substantial general-performance degradation.

### 3.4 INSIGHTS AND NEXT STEPS

**Beyond Smaller Learning Rates.** Building on the empirical and theoretical analyses above, we have shown that using a smaller learning rate can mitigate degradation in general performance while still achieving strong target-domain performance. However, small learning rates cannot solve everything. First, although smaller learning rates greatly reduce the extent of general-performance degradation, they do not fully eliminate it in some cases (Fig. 1g). This suggests that further strategies are needed to suppress such degradation more effectively. Second, while smaller learning rates generally achieve domain performance close to that of larger ones, in certain cases the gap is not entirely negligible (Fig. 1f and 1h). In situations where stronger target-domain performance is prioritized, larger learning rates may therefore be necessary, but they inevitably incur greater general-performance degradation. This makes the development of additional mitigation strategies under larger learning rates equally important in certain cases.

**Insights from Theoretical Analysis.** In Theorem 3.1, we can further expand the coefficients as

$$k_1 = \Theta(w_{\mathcal{S}} M_h + M_e), \qquad k_2 = \Theta(w_{\mathcal{S}} M_h + M_e + k_3),$$

where $M_h$ bounds the update magnitude on *hard* (low-probability) tokens, $M_e$ corresponds to easy tokens (with $M_h \gg M_e$), $w_{\mathcal{S}}$ denotes the mass of the hard-token set $\mathcal{S}$, and $k_3$ residual constants. For a fixed target dataset (hence essentially fixed $w_{\mathcal{S}}$), the dominant factor in both $k_1$ and $k_2$ is $M_h$. Therefore, *reducing* $M_h$, i.e., shrinking the update amplitude induced by low-probability (hard) tokens, tightens the upper bound on $\Delta L(P_1)$. This observation naturally motivates token-adaptive reweighting strategies that directly down-weight hard-token losses to curb their potential disproportionate influence to general performance degradation.

## 4 TOKEN-ADAPTIVE LOSS REWEIGHTING FOR DOMAIN-SPECIFIC SFT

From the preceding analysis, we see that reducing the update magnitude on low-probability tokens (hard tokens) can tighten the upper bound on general-performance degradation $\Delta L(P_1)$. This suggests a promising direction: down-weighting the loss of hard tokens to curb their disproportionate impact on forgetting. However, this immediately raises several practical challenges. How should we

Table 2: Comparison of domain and general performance on the MedCalc Benchmark under learning rate 5e−6. At this larger learning rate, TALR achieves the best overall trade-off by substantially improving general performance while maintaining comparable domain performance.

| Method | Qwen2.5-3B | | Qwen3-4B | | Gemma3-4B | | Average | |
|---|---|---|---|---|---|---|---|---|
| | Domain | General | Domain | General | Domain | General | Domain | General |
| Standard | 0.5459 | 0.3337 | 0.5782 | 0.5425 | 0.5507 | 0.2655 | 0.5583 | 0.3805 |
| L2-Reg | 0.5406 | 0.3470 | 0.5782 | 0.5591 | 0.5471 | 0.2796 | 0.5553 | 0.3952 |
| LoRA | 0.1734 | 0.5670 | 0.2367 | 0.7571 | 0.3864 | 0.1241 | 0.2655 | 0.4827 |
| Wise-FT | 0.3584 | 0.5869 | 0.3815 | 0.7531 | 0.4638 | 0.5929 | 0.4012 | 0.6443 |
| FLOW | 0.5266 | 0.4419 | 0.5819 | 0.5599 | 0.5500 | 0.3476 | 0.5528 | 0.4498 |
| **TALR (Ours)** | 0.5066 | 0.5490 | 0.5834 | 0.6138 | 0.5351 | 0.3427 | 0.5417 | 0.5018 |

identify which tokens are "hard"? If we rely on a fixed probability threshold, what value should be chosen? Even after identifying hard tokens, by how much should their losses be down-weighted? Manually setting such thresholds or scaling factors is cumbersome. To address these challenges, we propose a principled and adaptive solution, **TALR** (**T**oken-**A**daptive **L**oss **R**eweighting), to adaptively scales the loss contribution of each token according to its predicted probability. Additional details and discussions of TALR can be found in Appendix D.

### 4.1 Token-Adaptive Weight Computation via Constrained Optimization

Formally, let $\ell_i(\theta) = -\log p_\theta(x_i)$ denote the loss of token $i$ given model parameters $\theta$. We seek per-token weights $\mathbf{w} = (w_1, \ldots, w_n)$ that (1) assign smaller weights to harder tokens (*loss larger/probability lower $\Rightarrow$ weight smaller*); and (2) avoid collapsing all weight onto a small subset of tokens, ensuring broader coverage across the sequence. We formulate this as the following constrained optimization problem:

$$\min_{\mathbf{w} \in \Delta_n} \sum_{i=1}^{n} w_i \cdot \ell_i(\theta) + \tau \sum_{i=1}^{n} w_i \log w_i, \tag{1}$$

where $\Delta_n$ is the $n$-dimensional simplex ($w_i \geq 0$, $\sum_{i=1}^{n} w_i = 1$), and $\tau > 0$ controls the strength of entropy regularization. The first term enforces preference for low-loss tokens, while the negative-entropy regularization term prevents the distribution from becoming overly concentrated.

This optimization admits a closed-form solution: $w_i^* = \exp\left(-\ell_i(\theta)/\tau\right)/Z$, where $Z$ is the normalization factor. Since $\ell_i(\theta) = -\log p_\theta(x_i)$, we can equivalently write: $w_i^* \propto p_\theta(x_i)^{1/\tau}$.

In practice, we use the unnormalized form $w_i = p_\theta(x_i)^{1/\tau}$, focusing on the relative magnitudes. This also keeps $w_i \in (0, 1)$ naturally bounded and directly tied to the model's confidence. By scaling token-level loss with these adaptive weights, TALR tempers the excessive gradient contributions from low-probability tokens while preserving their influence for learning domain-specific knowledge. During training, these weights are recomputed at every optimization step for the tokens in the current batch, ensuring that the reweighting adapts dynamically to the model's evolving predictions. The detailed procedure is summarized in Algorithm 1.

### 4.2 Results

We evaluate all mitigation strategies considered in Sanyal et al. (2025), including L2 regularization, LoRA, Wise-FT (model averaging), and FLOW, together with our proposed TALR. Table 1 and 2 reports the trade-off between domain performance and general performance under two learning rates. The baseline configurations follow Sanyal et al. (2025).

**Smaller learning rate** (1e−6). From Table 1, most strategies, i.e., except LoRA and Wise-FT, achieve domain performance and general performance that are relatively close to each other. This indicates that simply using a small learning rate already mitigates the degradation of general capabilities while maintaining strong domain performance. Among all methods, both our Standard SFT (with smaller learning rates) and our TALR consistently provide the best trade-offs.

**Larger learning rate** (5e−6). From Table 2, we first observe that raising the learning rate amplifies general-performance degradation across nearly all methods. In this more challenging regime, TALR

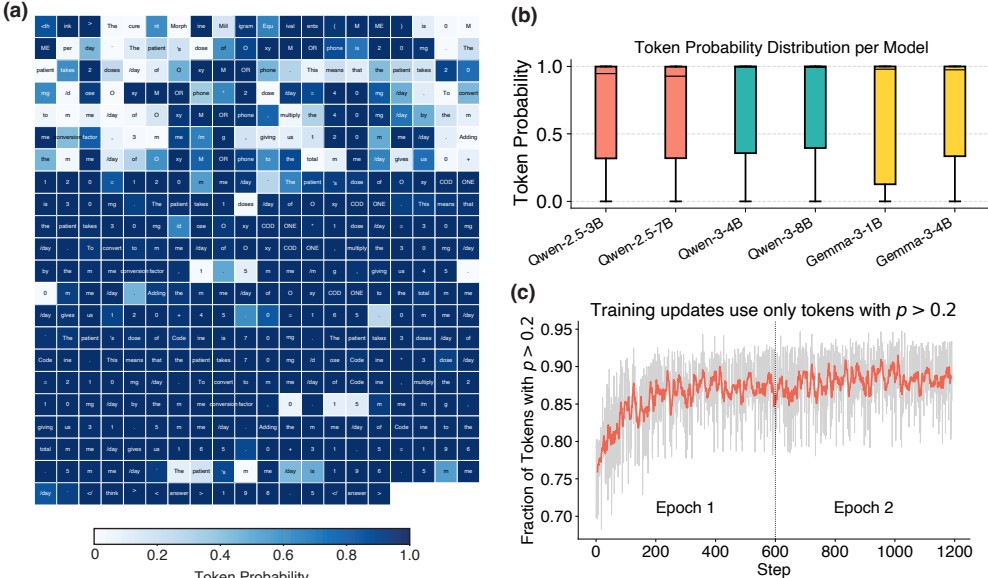

Figure 2: **Token-level analysis on the MedCalc dataset.** (a) Heatmap of token probabilities from Qwen-2.5-3B-Instruct for an example. Darker cells indicate higher model confidence; harder tokens with low probability often correspond to domain-specific concepts. (b) Distribution of token probabilities across the full SFT training set for multiple models. Most tokens are confidently predicted (medians near 1.0), suggesting low learning difficulty. (c) Fraction of tokens with $p > 0.2$ increases from epoch 1 to epoch 2 when training updates use tokens with $p > 0.2$, showing a clear curriculum phenomenon.

stands out: it achieves a clearly more favorable Pareto-optimal trade-off, maintaining competitive domain gains with noticeably smaller drops in general performance.

**Takeaway.** When feasible, a small learning rate already delivers a solid trade-off; additional knobs can be unnecessary. When higher learning rates are required to push domain performance, TALR demonstrates clear superiority by achieving stronger trade-offs. However, none of the existing methods, including TALR, can fully mitigate the sharp increase in general-performance degradation under larger learning rates. This highlights an open challenge and points to the need for further exploration of more powerful mitigation strategies.

### 4.3 TOKEN-LEVEL ANALYSIS

In this part, we analyze domain-specific SFT at a fine-grained level, i.e., at the level of individual target tokens. To this end, we compute the probability of each target token $x_t$ during SFT, where the model is trained to predict the next token conditioned on the prompt $x_{\text{prompt}}$ and previous target tokens $x_{<t}$, formulated as $p(x_t \mid x_{\text{prompt}}, x_{<t})$. This formulation allows us to quantify token difficulty.

**Finding 1: Most tokens in SFT training data pose low learning difficulty.** We begin our analysis with a token-level visualization of a training example from the MedCalc dataset. Figure 2(a) shows the model's predicted probability for each target token conditioned on the input prompt and all previous target tokens. Tokens with darker colors indicate higher confidence (i.e., higher probability or lower token loss), while lighter colors highlight tokens that the model finds more difficult. As shown, **the majority of tokens in the target sequence are confidently predicted by the model**, particularly in the later steps of the reasoning process. This aligns with the intuition that once sufficient context is accumulated, a well-trained LLM can easily predict subsequent tokens.

Notably, a small number of hard tokens, i.e., those with low predicted probabilities, do appear throughout the sequence, typically in earlier positions or around domain-specific concepts that may not be well covered in pretraining. For example, in the sixth row of the heatmap in Figure 2(a), the token representing the numeric value in the phrase "*conversion factor, 3 mme/mg*" is assigned a

---

**Input:** Domain dataset $\mathcal{D}$, parameters $\theta$, learning rate $\eta$, temperature $\tau > 0$, weight floor $w_{\min}$
**Output:** Updated parameters $\theta$

**foreach** *training step* **do**

    Sample a mini-batch $\{(x_{\text{prompt}}^{(b)}, y^{(b)})\}_{b=1}^{B}$ from $\mathcal{D}$;

    Forward pass to obtain token probabilities $\{p_t\}$ for all supervised tokens in the batch;

    Token NLLs: $\ell_t \leftarrow -\log p_t$;

    **Adaptive weights with lower-bound clipping:**

$$\tilde{w}_t \leftarrow \exp(-\ell_t/\tau), \qquad w_t \leftarrow \max\big( \text{sg}(\tilde{w}_t),\, w_{\min}\big)$$

    Let $N$ be the number of supervised tokens in the batch;

    **Mean (averaged) reweighted loss:**

$$\mathcal{L}_{\text{TALR}} \;=\; \frac{1}{N} \sum_{t=1}^{N} w_t \, (-\log p_t)$$

    Parameter update:

$$\theta \leftarrow \theta \;-\; \eta\, \nabla_\theta \mathcal{L}_{\text{TALR}}$$

**end**

---

**Algorithm 1:** Token-Adaptive Loss Reweighting (TALR) for Domain-Specific SFT. The $\text{sg}(\cdot)$ operator denotes *stop gradient*, meaning that $w_t$ is treated as a constant during backpropagation to prevent gradients from flowing through the weight computation.

low probability, likely because such clinical conversion factors are underrepresented in the model's pretraining data.

To move beyond a single example, we perform a broader statistical analysis by collecting token-level probabilities across all SFT data in the MedCalc training set. Figure 2(b) presents box plots of these token probabilities across six model variants. Across all models, we observe a consistent pattern: **the upper quartiles are tightly clustered near 1.0, and the medians are consistently high**, indicating that a large portion of tokens in the training sequences are already assigned high confidence by the models. However, despite this abundance of easy tokens, the models' zero-shot performance on the MedCalc test set remains relatively low, as shown in Figure 1 (*Init* point). This mismatch suggests that performance bottlenecks may stem from a small subset of more challenging tokens which are associated with domain-specific reasoning or clinical knowledge. These hard tokens may be sparse but crucial.

**Finding 2: TALR training dynamics exhibit a curriculum-like phenomenon.** We conducted an extreme experiment and observed that TALR implicitly creates a training curriculum. Specifically, we clipped the gradients of all tokens whose predicted probability was below a threshold, so that only higher-confidence tokens contributed to updates. As shown in Figure 2(c), the fraction of tokens exceeding this threshold grows steadily from Epoch 1 to Epoch 2. This dynamic effectively induces a *curriculum-like learning schedule*: the model begins with "easier" tokens (those already predicted with moderate confidence) and gradually incorporates a larger set of tokens, including those that were harder at the start.

## 4.4 Learning Dynamics of TALR

To analyze the actual behavior of TALR during optimization, we monitor several training-time signals for Qwen2.5-3B fine-tuned on the MedCalc dataset. Specifically, we track: (1) the token-level loss before reweighting, (2) the overall training loss after the TALR reweighting is applied, and (3) the dynamic hyperparameter $\tau$, defined as the median average token loss within each batch.

The results are shown in Figure 3. During the first epoch, both the token loss and the final training loss decrease sharply, and they continue to stabilize during the second epoch. Importantly, $\tau$ also decreases substantially as training progresses. Since $\tau$ reflects the median difficulty level of tokens within a batch, its steady decline indicates that a growing proportion of tokens transition from being initially hard to being easier for the model.

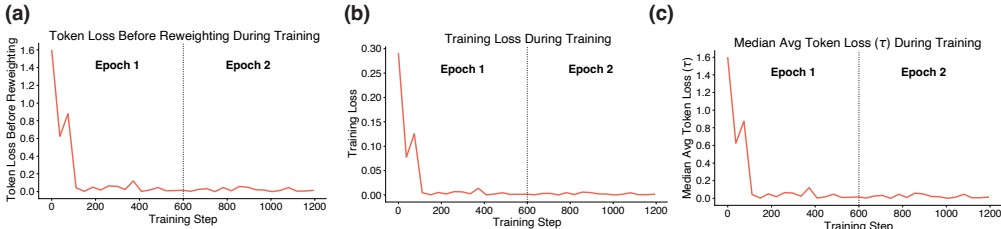

Figure 3: Training dynamics under the true TALR algorithm for Qwen2.5-3B-Instruct on MedCalc. Panel (a) shows the token-level loss before reweighting, panel (b) shows the training loss of TALR and panel (c) tracks the value of the dynamic hyperparameter $\tau$ (median average token loss) throughout training.

These observations confirm that the TALR reweighting mechanism does not impede learning. Instead, TALR allows the model to follow a normal optimization trajectory in which hard tokens are gradually absorbed, while simultaneously reducing the destabilizing influence of extremely low-probability tokens in early training.

## 5 CONCLUSION AND OUTLOOK

In this work, we presented both empirical and theoretical evidence that challenges the common belief that domain-specific SFT significantly harms general-purpose capabilities of LLMs. Through controlled experiments, we showed that smaller learning rates yield more favorable trade-offs. Motivated by our theoretical analysis, we further propose TALR for better trade-off.

### 5.1 LIMITATIONS

Looking forward, while TALR marks a step toward mitigating general-performance degradation in domain-specific adaptation, our findings also highlight that no single method fully resolves this challenge. Future work should explore more principled strategies to further enhance the robustness of LLMs across domains while preserving their general-purpose strengths. Second, due to no longer having access to compute resources for this project, we were not able to evaluate these representative mitigation strategies on a wider range of datasets. Nevertheless, our experiments provide consistent evidence supporting our main findings, and we leave it to the broader community to further examine and verify their generality. In addition, due to resource constraints, we were unable to examine whether larger models or mixture-of-experts (MoE) architectures follow the same dynamics, leaving open questions about scalability and architectural differences. Besides, on the theoretical side, while our analysis explains the observations, we did not address the problem of *how to optimally select* a learning rate that achieves the best trade-off in practice. Developing such principled selection rules remains an important direction for future work.

### 5.2 BROADER IMPACTS

**Better domain adaptation.** Our findings provide practitioners with insights when developing domain-specific LLMs. Taking the medical domain as an example, Jeong et al. (2024) show that existing medical-specialized LLMs often fail to outperform their corresponding initialized LLMs. This suggests that the quality of domain-specific data alone may not be as high as in the sophisticated post-training pipelines applied to base models. Hence, methods that preserve as much of the initialized LLM's general capabilities as possible while injecting domain knowledge may lead to stronger overall performance.

**Mitigating exploration loss in SFT warm-up for RLVR.** Before reinforcement learning with verifiable reward (RLVR), SFT is often used as a warm-up step to inject knowledge (Lin et al., 2025b) or align formats (Guo et al., 2025). However, excessive SFT can over-stabilize the model, causing its output trajectories to become rigid and thereby undermining exploration during RL training. In contrast, models prior to excessive SFT typically exhibit more diverse behaviors. Thus, strategies that mitigate general-performance degradation and preserve the base model's diversity may help alleviate this issue and enable more effective RL.

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

**Contents of Appendix**

## A    LLM Usage Statement

In this work, Large Language Models (LLMs) were primarily used for text refinement, such as improving the clarity of writing. In addition, for the ESCI experiments, the chain-of-thought (CoT) data was generated using Qwen2.5-72B-Instruct through rejection sampling.

## B    Theoretical Analysis

To better understand the empirical phenomena observed in §3, we provide a theoretical analysis from the perspective of information theory. Our goal is to explain **Finding 1** and **2** mentioned in §3.2. To this end, we first introduce several compression-based tools that form the basis of our analysis. We then apply these tools to shed light on the two key findings highlighted earlier.

### B.1    Preliminaries

**Supervised Fine-tuning (SFT).** In SFT, the LLM is trained on a labeled dataset $\mathcal{D}_{\text{SFT}} = \{(x^{(i)}, y^{(i)})\}_{i=1}^{N}$, where $x$ is a natural language prompt (e.g., an instruction or question), and $y = (y_1, y_2, \ldots, y_{T_y})$ is the corresponding target response, represented as a sequence of tokens. The objective of SFT is to maximize the conditional likelihood of the target sequence $y$ given the input $x$, which corresponds to minimizing the following negative log-likelihood loss: $\mathcal{L}_{\text{SFT}}(\theta) = -\mathbb{E}_{(x,y)\sim\mathcal{D}_{\text{SFT}}}\left[\sum_{t=1}^{T_y} \log \pi_\theta(y_t \mid x, y_{<t})\right]$, where $\pi_\theta$ denotes the model's output distribution over the vocabulary, and $y_{<t}$ is previous target tokens.

**Lossless Compression.** In lossless compression, the goal is to encode a sequence of symbols $x = (x_1, x_2, \ldots, x_T)$ drawn from a source distribution $P$ into a binary representation without any loss of information, such that the original sequence can be perfectly reconstructed. According to Shannon's source coding theorem (Shannon, 1948), the limit of compression is given by the Shannon entropy of the source: $H(P) := \mathbb{E}_{x\sim P}\left[-\log P(x)\right]$, which specifies the minimum expected number of bits per symbol needed for encoding.

**LLM Modeling is Compression.** Given a dataset $\mathcal{D}$ drawn from the true distribution $P$ and a model distribution $Q$, the expected code length under arithmetic coding (Witten et al., 1987) is given by $H(P, Q) := \mathbb{E}_{x\sim P}\left[-\log Q(x)\right]$. Thus, minimizing the log-likelihood loss directly corresponds to reducing the expected compression rate when the model is employed as a lossless compressor (Deletang et al., 2024; Hutter, 2005; Ji et al., 2025).

### B.2    LLM Compression Protocol

Our goal is to analyze the dynamics of domain-specific SFT. Motivated by the equivalence between language modeling and data compression (Deletang et al., 2024; Ji et al., 2025), we view an LLM as a *compressor*, where the effectiveness of training can be measured through changes in code length. In this view, improvements or degradations in performance across datasets correspond to variations in compression rate. Below, we formalize this perspective by introducing the notion of token trees and describing the LLM compression protocol in our context.

> **Definition B.1** (Token Tree $\mathcal{T}$). *For a dataset $\mathcal{D} = \{z_i \in \mathcal{V}^\infty \mid i = 1, 2, \ldots\}$, $|\mathcal{V}| < \infty$, where $\mathcal{V} = \{v_1, v_2, \ldots, v_{|\mathcal{V}|}\}$ is a finite vocabulary of size $|\mathcal{V}|$, the token tree of $\mathcal{D}$, denoted as $\mathcal{T}_\mathcal{D}$, is defined as follows: (1) each node has $|\mathcal{V}|$ child nodes labeled $v_1, v_2, \ldots, v_{|\mathcal{V}|}$, along with an end-of-sequence (EOS) leaf node; (2) The weight of a non-leaf node is the sum of the weights of all its child nodes; (3) The path from the root to an EOS leaf node defines a response $z_i$, with the corresponding EOS node weight representing the response's probability.*

> **Definition B.2** (LLM Compression Protocol). *Let $\mathcal{T}_\mathcal{D}$ be the token tree of dataset $\mathcal{D}$, and let $q_\theta(\cdot \mid u)$ denote the conditional distribution over $\mathcal{V} \cup \{\text{EOS}\}$ predicted by an LLM with parameters $\theta$ at node $u \in \mathcal{T}_\mathcal{D}$. Given a response $z$ (a path from the root to an EOS leaf, truncated to a predefined maximum depth $d$), the LLM compression protocol encodes $z$ using* arithmetic coding, *where at each step the coding probabilities are given by $q_\theta(\cdot \mid u)$ for the current node $u$ along the path of $z$.*

**Remark:** The truncation to a maximum depth $d$ reflects practical constraints in the use of large language models. For example, responses are usually limited to a fixed context window, and generated sequences are typically bounded in length.

**Proposition B.1** (Expected Code Length). *Consider a finite parameter model $q_\theta(\cdot)$ and a token tree $\mathcal{T}_\mathcal{D}$ truncated to depth $d$. Under the compression protocol of Definition 3.2, the expected code length of a random response $z$ is $\mathbb{E}_{z \sim P}[L_\theta(z)] = -\sum_{l=1}^{d} \sum_{j=1}^{|\mathcal{V}|^{l-1}} p_{l,j} \log q_{l,j}$, where $P$ is the distribution over responses, $p_{l,j}$ denotes the probability assigned to the leaf node $u_{l,j}$ (the $j$-th node at layer $l$ of $\mathcal{T}_\mathcal{D}$), and $q_{l,j}$ is the probability assigned to the node $u_{l,j}$ by the model $q_\theta(\cdot)$.*

**Proposition B.2** (Joint Token Tree for Multiple Datasets). *Consider $N$ pairwise disjoint datasets $\mathcal{D}_1, \dots, \mathcal{D}_N$, each with its own token tree $\mathcal{T}_{\mathcal{D}_i}$. Let $\mathcal{D} = \bigcup_{i=1}^{N} \mathcal{D}_i$ be the union dataset, and let $\mathcal{T}_\mathcal{D}$ denote its token tree. For each node $u_{l,j}$, the node weight in $\mathcal{T}_\mathcal{D}$ is given by $p_{l,j}^{\mathcal{D}} = \left(\sum_{i=1}^{N} |\mathcal{D}_i| p_{l,j}^{\mathcal{D}_i}\right) / \left(\sum_{i=1}^{N} |\mathcal{D}_i|\right)$, where $p_{l,j}^{\mathcal{D}_i}$ is the node weight in $\mathcal{T}_{\mathcal{D}_i}$, and $|\mathcal{D}_i|$ is the number of responses in dataset $\mathcal{D}_i$.*

**Proposition B.3** (Expected Code Length Discrepancy under Model Shift). *Consider two model distributions $q_{\theta_1}(\cdot)$ and $q_{\theta_2}(\cdot)$ over the token tree $\mathcal{T}_\mathcal{D}$ with distribution $P$. The change in expected code length on $P$ when shifting from $q_{\theta_1}$ to $q_{\theta_2}$ is $\Delta L(P) = \mathbb{E}_{z \sim P}[L_{q_{\theta_2}}(z)] - \mathbb{E}_{z \sim P}[L_{q_{\theta_1}}(z)] = -\sum_{l=1}^{d} \sum_{j} p_{l,j} \log \frac{q_{l,j}^{(2)}}{q_{l,j}^{(1)}}$. Equivalently, $\Delta L(P) = \text{KL}(P \,\|\, q_{\theta_2}) - \text{KL}(P \,\|\, q_{\theta_1})$.*

Based on the above, we adopt the expected code length as a surrogate metric for an LLM's modeling quality on a given dataset (Deletang et al., 2024). Specifically, reductions in code length discrepancy indicate better alignment between the model distribution and the data distribution, whereas increases suggest deterioration. This perspective will serve as the foundation for our subsequent analysis.

### B.3 Approximating Fine-tuning Dynamics with Exponential Tilting

Our goal in this part is to uncover the behavior of LLMs during domain-specific fine-tuning. ***Fine-tuning alters the conditional distributions assigned to each node in the token tree***, thereby shifting the model's alignment with the data distribution. For analytical simplicity, we view fine-tuning at a high level as introducing perturbations to the probability assigned to each token node. To approximate these dynamics, we adopt the lens of *exponential tilting* (Maity et al., 2023), which captures how the distribution is reweighted under incremental updates.

Note that, exponential tilting is not strictly equivalent to SFT; rather, it serves as an **analytical surrogate that enables us to extract insights and motivation about the mechanisms** driving general-performance degradation and domain adaptation. In the following, we formalize the exponential tilting formulation and present how it approximates the token-level probability shifts induced by fine-tuning. We then provide error estimates that quantify the gap.

**Setup.** We consider a pretrained LLM with distribution $q_0$, which already models the dataset $\mathcal{D}_1$ well. The model is then fine-tuned on a new dataset $\mathcal{D}_2$. For a node $u$ in the token tree $\mathcal{T}_{\mathcal{D}_2}$ of $\mathcal{D}_2$, let $\hat{p}_2(\cdot \mid u)$ denote the empirical target distribution induced by $\mathcal{D}_2$, and let $q_t(\cdot \mid u)$ denote the model distribution at step $t$ during fine-tuning.

**Assumption B.1** (Full support via mild smoothing). *To avoid support collapse, the target used in each step is defined as a smoothed mixture $\tilde{p}_{2,t}(\cdot \mid u) = (1 - \alpha)\hat{p}_2(\cdot \mid u) + \alpha \rho_t(\cdot \mid u), \alpha \in (0, 1)$, where $\rho_t(\cdot \mid u)$ is a strictly positive reference distribution (e.g., the current model $q_t(\cdot \mid u)$ or the uniform distribution). Hence $\tilde{p}_{2,t}(a \mid u) > 0$ and $q_t(a \mid u) > 0$ for all tokens $a$.*

**Assumption B.2** (Small step in distribution space). *Each update is small at the distribution level: $\text{KL}(q_t \,\|\, q_{t+1}) \leq \varepsilon, \varepsilon \ll 1$.*

**Assumption B.3** (Smoothness / finite tree). $\log q_\theta(a \mid u)$ *is twice continuously differentiable in $\theta$ with bounded second derivatives in a neighborhood of $\theta_t$; vocabulary and depth are finite. Consequently, Taylor remainders are $O(\|\theta_{t+1} - \theta_t\|^2)$.*

---

**Definition B.3** (Exponential Tilting Update). *For any non-leaf prefix $u$ in the token tree $\mathcal{T}$ and a step parameter $\lambda \in [0,1]$, define the log-ratio $r_u(a) \triangleq \log\big(\tilde{p}_{2,t}(a \mid u)/q_t(a \mid u)\big), a \in \mathcal{V} \cup \{\text{EOS}\}$. The* exponential tilting update *at prefix $u$ is given by*

$$\widehat{q}_{t+1}(a \mid u) = \frac{q_t(a \mid u)\,\exp\{\lambda\,r_u(a)\}}{\sum_b q_t(b \mid u)\,\exp\{\lambda\,r_u(b)\}} = \frac{q_t(a \mid u)^{1-\lambda}\,\tilde{p}_{2,t}(a \mid u)^\lambda}{\sum_b q_t(b \mid u)^{1-\lambda}\,\tilde{p}_{2,t}(b \mid u)^\lambda}.$$

*The boundary cases are consistent: $\lambda = 0$ recovers $q_t(\cdot \mid u)$, while $\lambda = 1$ recovers $\tilde{p}_{2,t}(\cdot \mid u)$.*

---

**Theorem B.1** (First-order approximation by exponential tilting). *Fix a prefix $u$. Consider the current model distribution $q_t(\cdot \mid u)$ and the smoothed target distribution $\tilde{p}_{2,t}(\cdot \mid u)$. Define the local $L^2$ norm $\|g\|_{t,u} := \big(\mathbb{E}_{q_t(\cdot\mid u)}[g(a)^2]\big)^{1/2}$. Under the standing assumptions, there exists an effective step size $\lambda_{t,u}$, such that*

$$\left\|\log q_{t+1}(\cdot \mid u) - \Big[(1 - \lambda_{t,u})\log q_t(\cdot \mid u) + \lambda_{t,u}\log\tilde{p}_{2,t}(\cdot \mid u) - \psi_{t,u}\Big]\right\|_{t,u} = O(\varepsilon).$$

*where $\psi_{t,u}$ is the log-normalizer and $\varepsilon$ is the KL trust-region radius such that $\mathrm{KL}\big(q_t \,\|\, q_{t+1}\big) \leq \varepsilon$.*

---

The proof can be found in §E.1. Based on Theorem B.1, we establish that for a distribution update $q_{t+1}$ whose KL divergence from $q_t$ is bounded by $\epsilon$, the corresponding exponential-tilting approximation $\widehat{q}_{t+1}$ differs from $q_{t+1}$ only up to $O(\epsilon)$. In other words, exponential tilting provides a first-order approximation, thereby justifying its use as an analytical tool to study general-performance degradation and domain adaptation.

## B.4 WHY SMALLER LEARNING RATES YIELD FAVORABLE TRADE-OFFS?

In this subsection, we provide a theoretical explanation for the empirical findings observed in §3.

**Notation.** Fix a prefix $u$ (we omit "$\mid u$" when clear). Write $f(a) = \log\tilde{p}_2(a) - \log q(a)$ at the current iterate $q$ (the step index $t$ is omitted for readability), and $\bar{f} = \mathbb{E}_q[f]$, $\tilde{f} = f - \bar{f}$. For a set $\mathcal{S}$ of token-tree nodes, denote its $q$-mass by $w_{\mathcal{S}} = \mathbb{E}_q[\mathbf{1}_{\mathcal{S}}]$.

**Assumption B.4** (Sparse token-level shift on $\mathcal{D}_2$). *There exists a measurable node set $\mathcal{S} \subseteq \mathcal{T}$ with small mass $w_{\mathcal{S}} \ll 1$ under the $\mathcal{D}_2$-prefix distribution such that*

$$|f(a)| \leq M_h \quad \text{for } a \in \mathcal{S}, \qquad |f(a)| \leq M_l \quad \text{for } a \notin \mathcal{S},$$

*with $M_l \ll M_h$. In words, most tokens are already well modeled by $Q$ due to pretraining, while only a small subset requires nontrivial adjustment toward the $\tilde{p}_2$ target, denoted as the hard tokens (low probability tokens).*

**Remark.** This assumption reflects the practical setting where domain-specific finetuning are mainly affected by a small fraction of tokens, consistent with our empirical analysis in §4.3.

**Assumption B.5** (Realizability with controlled leakage). *Let $\Pi_{\mathcal{S}}$ and $\Pi_{\mathcal{S}^c}$ denote the orthogonal projections of a log-space function onto the coordinates indexed by $\mathcal{S}$ and its complement, respectively (with norm measured in $L^2(q)$). There exists small $\gamma \in [0,1)$ such that for any desired log-space direction $g$ (defined per-prefix on the token tree) whose energy is primarily on $\mathcal{S}$ (i.e., $\|\Pi_{\mathcal{S}^c}g\|_{L^2(q)} \leq \beta\,\|\Pi_{\mathcal{S}}g\|_{L^2(q)}$ for some small $\beta \geq 0$), there is a parameter update that realizes a global change $\Delta\log q$ satisfying*

$$\|\Pi_{\mathcal{S}}(\Delta\log q - g)\|_{L^2(q)} \leq \gamma\,\|\Pi_{\mathcal{S}}g\|_{L^2(q)}, \qquad \|\Pi_{\mathcal{S}^c}\Delta\log q\|_{L^2(q)} \leq (\beta + \gamma)\,\|\Pi_{\mathcal{S}}g\|_{L^2(q)}.$$

*In words, the update moves all tokens, but the relative magnitude outside $\mathcal{S}$ is small and controlled.*

**Theorem B.2** (Smaller steps yield a smaller general performance degradation bound at a equal domain performance gain). *Fix a desired domain improvement $\Delta_\star > 0$ on $\mathcal{D}_2$ (i.e., $\Delta L_T(P_2) \leq -\Delta_\star$). Among all $T$-step tilting schedules that achieve this target, the* minimal *upper bound on the increase of code length on $\mathcal{D}_1$ satisfies*

$$\Delta L_T(P_1) \leq A\frac{\Delta_\star}{\mu_T} + \Big(\frac{A\,C_2}{\mu_T^3} + \frac{C_1}{\mu_T^2}\Big)\frac{\Delta_\star^2}{T} + O\Big(\frac{1}{T^2}\Big)$$

*where $\mu_T := \inf_{t<T} \mathrm{KL}(Q_t\|P_2) > 0$ and $A := H_T\big(\sqrt{w_{\mathcal{S}}}\,M_h + M_l + (\beta+\gamma)M_h\big)$ are fixed value under the total number of update steps $T$ and the desired domain gain $\Delta_\star$.*
*The upper bound strictly decreases as $T$ increases. Thus, under the equal-steps schedule that attains the target, the per-step effective weight scales as $\lambda_t \propto 1/T$; thus, for the same domain gain, larger $T$ implies smaller per-step updates. Hence, smaller step size $\Rightarrow$ smaller upper bound.*

**Theorem B.3** (Label-only supervision enlarges the safe per-step range). *Among all $T$-step tilting schedules, the maximal per-step size that can guarantee a general-performance degradation $\Delta L_T(P_1) \leq \varepsilon_{\mathrm{fg}}$ as $\lambda_{\max} = \Theta(1/\sqrt{s})$, where $s$ is the expected number of hard tokens (low probability tokens) per example on $\mathcal{D}_2$.*

The proof of Theorem B.2 and B.3 can be found in the Appendix E.2. Theorem B.2 shows that, for achieving the same domain improvement, smaller learning rates (i.e., smaller per-step updates with larger $T$) lead to a smaller upper bound on general capability degradation, thereby explaining **Finding 1**. Theorem B.3 indicates that the bound on the safe step size is inversely proportional to the number of hard tokens. Therefore, in **Finding 2**, when only labels are used for training, the number of hard tokens is smaller than that in training with both CoT and label data. This explains why in the ESCI experiments, under w/o CoT, both $5e-6$ and $1e-6$ can achieve similarly small degradation in general performance.

## C  ADDITIONAL EXPERIMENT AND RESULT DETAILS

### C.1  DATASET DETAILS

In this section, we provide additional details for both the domain-specific datasets used for SFT and the general-purpose benchmarks used to evaluate general capability degradation. An overview of all datasets and their corresponding evaluation metrics is provided in Table 4 and Table 5.

#### C.1.1  MEDCALC

We use the MedCalc dataset (Khandekar et al., 2024) for medical reasoning tasks. The benchmark provides human-annotated chain-of-thought (CoT) rationales, which we include during training so that the model learns to reason through intermediate steps before producing the final answer. The prompt can be found in Table 6.

#### C.1.2  ESCI

We use the ESCI dataset (Reddy et al., 2022) for a multi-class product classification task, where each query–product pair is labeled as *Exact*, *Substitute*, *Complement*, or *Irrelevant*. From the original dataset, we randomly sample a 50K subset from the training split and a 10K subset from the test split. From the training subset, we further hold out 1K examples as a validation set.

We consider two training settings: *w/o CoT* and *w/ CoT*.

- **w/ CoT**: The target sequence includes both a chain-of-thought rationale and the final label, requiring the model to learn the reasoning process before producing the prediction. These CoT-augmented examples are generated via rejection sampling from Qwen2.5-72B-Instruct, resulting in 34,176 training examples. The prompt is shown in Table 8.
- **w/o CoT**: The target sequence contains only the ground-truth label, so the model is trained to directly predict the class without generating intermediate reasoning (49k examples). The prompt is shown in Table 7.

All prompt examples in Tables 7 and 8 use the Qwen chat template for illustration; for other model families, we adapt the prompt to their respective chat formats.

The ESCI dataset is highly imbalanced, with the majority of samples belonging to the *Exact* category (Table 3). This imbalance motivates our choice of *balanced accuracy* (BACC) as the primary evaluation metric, following prior work on imbalanced classification (Xu et al., 2024; 2025).

Table 3: Label distribution for the ESCI subsets used in our experiments. Percentages are shown in parentheses.

| Split | Exact | Substitute | Irrelevant | Complement |
|---|---|---|---|---|
| Train (49K) | 33,958 (69.30%) | 9,753 (19.90%) | 4,261 (8.70%) | 1,028 (2.10%) |
| Val (1K) | 674 (67.40%) | 212 (21.20%) | 94 (9.40%) | 20 (2.00%) |
| Test (10K) | 6,470 (64.70%) | 2,268 (22.68%) | 992 (9.92%) | 270 (2.70%) |

### C.1.3 METAMATHQA

MetaMathQA (Yu et al., 2024) is a large-scale mathematical reasoning dataset containing 395k training examples. Following Sanyal et al. (2025), we use MetaMathQA for training and take GSM8K as the target-domain evaluation benchmark. This setup allows us to validate whether our findings hold under large-scale data conditions.

### C.1.4 GENERAL-PURPOSE BENCHMARKS

For the general-purpose benchmarks, we fully follow the default settings and evaluation metrics implemented in the `lm-evaluation-harness` framework (Gao et al., 2024). This ensures consistency with prior work (Lin et al., 2025a; Sanyal et al., 2025; Bansal & Sanghavi, 2025) and allows for fair comparison of results across different models and training configurations.

## C.2 IMPLEMENTATION DETAILS

We conduct all experiments on 16–32 NVIDIA A100 GPUs with 80GB memory. Except for differences in learning rate and loss computation, all experiments share the same training configuration. We adopt the AdamW optimizer (Loshchilov & Hutter, 2019) with hyperparameters $\beta_1 = 0.9$ and $\beta_2 = 0.999$, together with a cosine annealing learning-rate schedule. The attention mechanism is implemented using FlashAttention-2 (Dao, 2024). We set the batch size to 16 for MedCalc and ESCI, and 128 for MetaMathQA. The number of training epochs is 20 for MedCalc and ESCI, and 2 for MetaMathQA. The maximum sequence length is 8192 tokens.

## C.3 ADDITIONAL DETAILS OF EXPERIMENTAL SETUP AND RESULTS

In our experiments, we measure the trade-off between domain performance and general performance. Domain performance is defined as accuracy on the target downstream task, while general performance is computed as the average score across IFEval, GSM8K, and HumanEval unless otherwise specified. Importantly, our definition of general performance is consistent with the theoretical analysis, where we assume the base model already achieves reasonably strong results. To ensure consistency, we exclude benchmarks where the model's absolute performance is below a threshold of 0.5, evaluated by `lm-evaluation-harness` framework. Thus, for Gemma-3-4B we report the average over IFEval and GSM8K, while for Gemma-3-1B we only include IFEval.

We also conduct supplementary experiments, as shown in Figure 4, which further validate and extend our findings from Section 3.

**Finetuning on datasets where the model already performs strongly.** In Figure 4(a), Qwen3-8B achieves close to 50% accuracy on ESCI with CoT supervision even before SFT. Despite this high baseline, the results confirm our main conclusion: using a small learning rate continues to yield a more favorable trade-off between preserving general performance and improving domain performance.

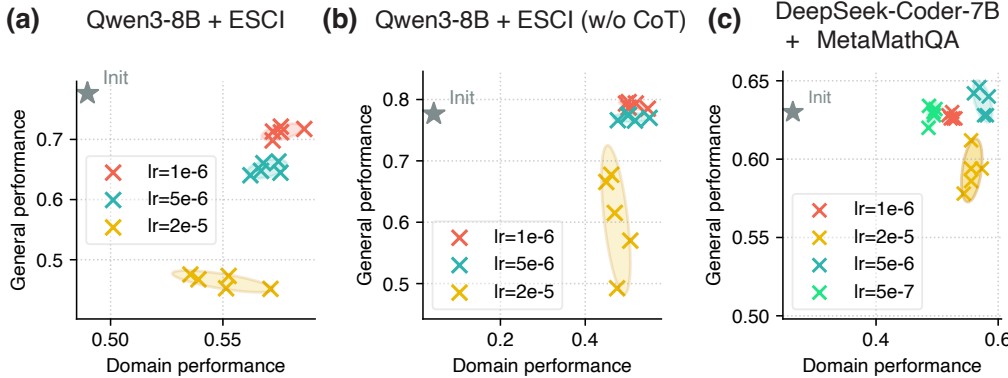

Figure 4: **Effect of learning rate on domain-specific and general capability performance during supervised fine-tuning (SFT).** Results are shown for (a) Qwen3-8B on ESCI with CoT supervision, (b) Qwen3-8B on ESCI without CoT, and (c) DeepSeek-Coder-7B on MetaMathQA. Across all settings, smaller learning rates achieve more favorable trade-offs.

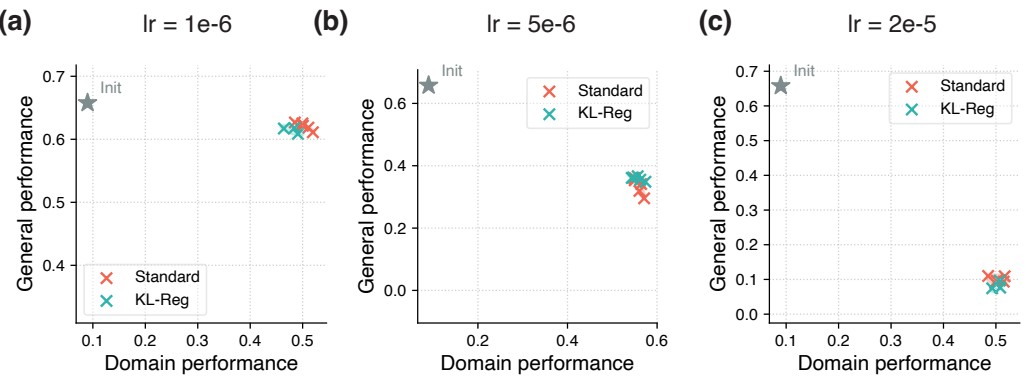

Figure 5: **Effect of KL regularization on domain-specific SFT.** We follow DeepSeek-R1 (Guo et al., 2025) and apply the $k3$ approximation for KL regularization. Results are shown for three learning rates: (a) $1 \times 10^{-6}$, (b) $5 \times 10^{-6}$, and (c) $2 \times 10^{-5}$. Across all settings, KL regularization yields performance that is very close to standard SFT, suggesting limited additional benefit in mitigating general-performance degradation.

**Validation on large-scale datasets.** We additionally evaluate on MetaMathQA to test whether our conclusions hold under large-scale training. To emulate a realistic domain adaptation scenario, we use DeepSeek-Coder-7B, which is highly specialized in code but weaker in mathematics. This setup mirrors adapting a model from one domain of strength (code) to another (math). As shown in Figure 4(c), we report general performance using MBPP (rather than HumanEval, since DeepSeek-Coder-7B performs poorly on HumanEval under `lm-evaluation-harness`). The results again align with our central finding: small learning rates achieve the best trade-offs. Interestingly, in this setting the optimal rate shifts to $5 \times 10^{-6}$, rather than $1 \times 10^{-6}$ as in earlier experiments. Moreover, we test an even smaller rate of $5 \times 10^{-7}$ and observe that **overly small rates can hinder target-domain performance**, suggesting that learning rates cannot be arbitrarily reduced without consequence. Overall, these additional experiments reinforce our main findings.

## C.4 EFFECT OF KL REGULARIZATION

We further investigate the effect of KL regularization, a technique recently adopted in DeepSeek-R1 (Guo et al., 2025), where a $k3$ approximation is used to estimate the KL term. Following prior work

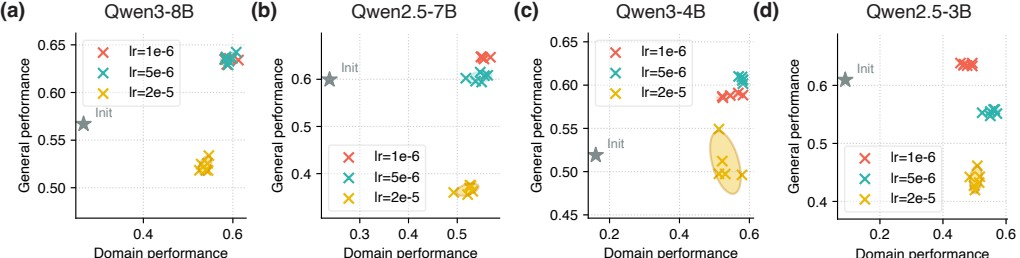

Figure 6: **Effect of learning rate on the trade-off between domain performance and general multi-choice commonsense and knowledge QA performance.** Domain performance is measured on MedCalc, while general performance is evaluated as the average accuracy across MMLU, ARC-Easy, ARC-Challenge, PIQA, and HellaSwag. Results are shown for (a) Qwen3-8B, (b) Qwen2.5-7B, (c) Qwen3-4B, and (d) Qwen2.5-3B.

on KL-constrained training (Jin et al., 2025; Jiang et al., 2025; Lin et al., 2025a;b), we add a KL penalty term with coefficient $0.001$ during SFT.

Figure 5 shows results on the Qwen2.5-3B-Instruct model fine-tuned on MedCalc. At small learning rates, KL-regularized runs and standard SFT behave almost identically. As the learning rate increases, KL regularization offering little to no benefit in reducing general-performance degradation. This indicates that, under our experimental settings, KL regularization provides only limited improvements and does not shift the trade-off between domain performance and general capability preservation. These results are consistent with our earlier observation in §3: adopting a smaller learning rate already achieves a favorable balance, while additional knobs such as KL regularization contribute little further advantage.

## C.5 EVALUATION ON MULTI-CHOICE COMMONSENSE AND KNOWLEDGE QA

We further evaluate the effect of learning rate on the trade-off between domain and general performance in multi-choice commonsense and knowledge question answering tasks. Results are presented in Figure 6. Unlike our earlier observations on more complex domains such as mathematics and coding, we find that the general-performance degradation induced by relatively larger learning rates (e.g., $5e{-}6$) is less pronounced here. A possible explanation is that multi-choice benchmarks are relatively more trivial, requiring short-form predictions rather than long reasoning chains or structured outputs. As a result, larger learning rates do not amplify forgetting as severely as in domains demanding longer and more complex generations.

## C.6 PERFORMANCE EVOLUTION ACROSS TRAINING EPOCHS

To better understand how learning rate influences the interaction between domain performance and general performance over the training process, we plot figures of performance vs. training epochs in Figure 7. Figure 7a and 7b report results for Gemma3-4B on the MedCalc benchmark. We make two observations. First, consistent with our main findings, the smallest learning rate achieves strong domain performance while substantially mitigating forgetting on general benchmarks. Second, for each learning rate, the domain performance typically peaks at relatively late epochs. For example, for Gemma3-4B with a learning rate of $1e{-}6$, the best MedCalc score is reached around epoch 12. This indicates that, during domain-specific SFT, running training for a longer period can continue to improve domain accuracy.

We further examine a larger scale setting with DeepSeek-Coder-7B on MetaMathQA, as shown in Figure 7c and 7d. We observe similar behavior. Domain performance improves steadily and often reaches its peak only after many optimization steps, and smaller learning rates help preserve general capabilities and achieve comparable or even better domain performance. Note that in this case the horizontal axis covers only the first two epochs, but since MetaMathQA is a large scale corpus, even one epoch already corresponds to a large number of parameter update steps. These results confirm

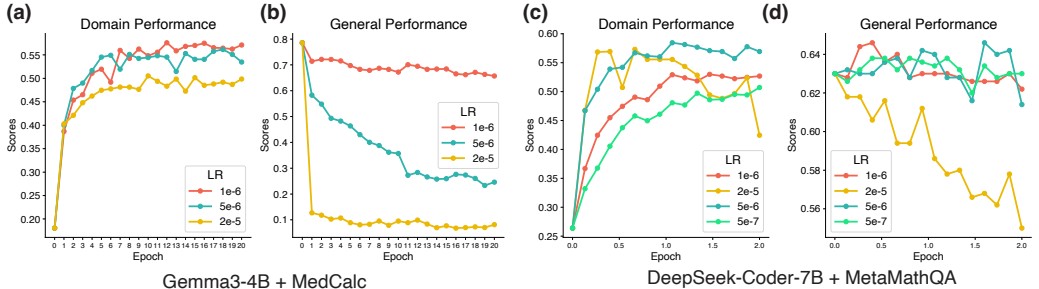

Figure 7: Training dynamics of domain and general performance under different learning rates. Panels (a) and (b) show domain performance and general performance respectively for Gemma3-4B on MedCalc. Panels (c) and (d) show the corresponding curves for DeepSeek-Coder-7B on MetaMathQA. In both settings, small learning rates achieve strong domain performance while better preserving general capabilities.

that our conclusions are not restricted to small datasets and that the dynamics we describe persist in a large scale training datasets.

## C.7  OBSERVING THE RATIO OF LOW-PROBABILITY TOKENS

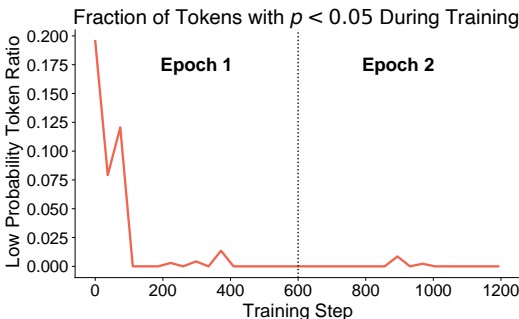

Figure 8: Fraction of low-probability tokens during training on Qwen2.5-3B-Instruct with Med-Calc. We track the proportion of tokens whose model probability satisfies $p < 0.05$ over the course of training. During the first epoch, the ratio of such "hard" tokens decreases rapidly, and by the second epoch it approaches zero and remains near zero. This indicates that these initially low-probability tokens are successfully learned by the model as training progresses.

To examine whether tokens assigned low probabilities are eventually learned during training, we track the evolution of the proportion of such tokens for Qwen2.5-3B-Instruct fine-tuned on the MedCalc dataset. We define low-probability tokens as those with predicted probability less than $p < 0.05$. Figure 8 plots the ratio of these tokens across training steps.

During the first epoch, the fraction of low-probability tokens decreases sharply, indicating that many of these hard tokens are quickly absorbed by the model. By the second epoch, this ratio approaches zero and remains near zero for the rest of training. This pattern shows that low-probability tokens do not remain persistently difficult; instead, they are gradually learned as training progresses. This analysis provides direct evidence that TALR does not prevent the model from learning challenging tokens.

## D    DETAILS OF TOKEN-ADAPTIVE LOSS REWEIGHTING

### D.1    DERIVING TOKEN WEIGHTS: PROOF OF THE CLOSED-FORM SOLUTION

*Proof.* Introduce a Lagrange multiplier $\lambda$ for the simplex constraint $\sum_i w_i = 1$, and multipliers $\mu_i \geq 0$ for the nonnegativity constraints. The Lagrangian is

$$\mathcal{L}(\mathbf{w}, \lambda, \boldsymbol{\mu}) = \sum_{i=1}^{n} \Big( w_i \ell_i(\theta) + \tau w_i \log w_i \Big) + \lambda \Big( \sum_{i=1}^{n} w_i - 1 \Big) - \sum_{i=1}^{n} \mu_i w_i.$$

For an interior optimum ($w_i > 0$ so that $\mu_i = 0$), the KKT condition is

$$\frac{\partial \mathcal{L}}{\partial w_i} = \ell_i(\theta) + \tau(1 + \log w_i) + \lambda = 0.$$

Thus,

$$\log w_i = -\frac{\ell_i(\theta) + \lambda}{\tau} - 1 \quad \implies \quad w_i = \exp\Big(-\frac{\ell_i(\theta)}{\tau}\Big) \cdot \exp\Big(-\frac{\lambda}{\tau} - 1\Big).$$

Normalization by $\sum_i w_i = 1$, then we have

$$Z = \sum_{j=1}^{n} \exp\Big(-\frac{\ell_j(\theta)}{\tau}\Big), \qquad w_i^* = \frac{\exp\big(-\ell_i(\theta)/\tau\big)}{Z}.$$

$\square$

### D.2    IMPLEMENTATION DETAILS OF TALR

We highlight two key design considerations in applying TALR.

**Weight cutoff.** Without constraints, hard tokens may receive extremely small weights, which slows down learning or even prevents the model from learning these tokens. To address this, we introduce a lower bound cutoff to ensure that no token weight becomes too small. In all our experiments, we set this cutoff to $0.01$, which strikes a balance between preventing vanishing weights and still allowing TALR to downweight challenging tokens.

**Choice of $\tau$.** The temperature $\tau$ controls the sharpness of weight assignment and is a crucial hyperparameter. In our experiments, $\tau$ is chosen dynamically as the median of the average sequence loss within a batch, a strategy that consistently yields stable and strong performance across tasks. To better illustrate the effect of $\tau$, we plot Figure 9. When a batch contains more hard tokens, the resulting $\tau$ is larger; in this case, weights assigned to hard tokens are not excessively small, preventing the model from failing to learn. Conversely, when the overall loss is smaller, the resulting $\tau$ decreases, which effectively acts as a hard clipping mechanism to prevent excessive parameter drift and catastrophic forgetting. Nonetheless, the problem of selecting $\tau$ remains open, and future work may explore more principled or adaptive strategies for temperature tuning in TALR.

### D.3    MORE DISCUSSIONS

**Comparison with FLOW.** It is worth contrasting TALR with FLOW (Sanyal et al., 2025), which also reweights losses but in a different manner. First, FLOW operates at the *sequence level*, whereas TALR works at the *token level*. Second, FLOW computes static weights only once before training, while TALR dynamically updates weights at every batch with negligible additional cost. As shown in Table 1 and Table 2, TALR consistently outperforms FLOW, which aligns with our expectations. Sequence-level loss can be misleading: for example, even when the overall average sequence loss is small, there may exist a few particularly hard tokens with large losses that are overlooked at the sequence granularity. TALR directly addresses this by reweighting at the token level. Moreover, token difficulty is not fixed—its relative hardness evolves as training progresses, as discussed in Section 4.3. This makes dynamic weighting naturally more advantageous than static approaches.

**Why not fix $\tau$.** We also examine the impact of fixing the temperature parameter, e.g., setting $\tau = 1$. In this case, the weights of hard tokens become excessively small, which severely hampers the model's ability to learn from them. Empirically, we observe that such a fixed choice indeed leads to

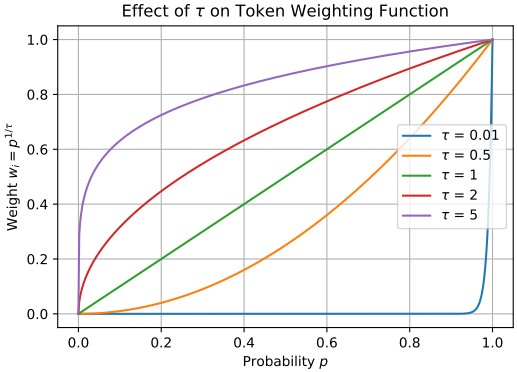

Figure 9: **Effect of the temperature parameter $\tau$ on the token weighting function** $w_i = p^{1/\tau}$. Smaller $\tau$ values (e.g., $\tau = 0.5$ or $\tau = 0.01$) sharply down-weight low-probability (hard) tokens, leading to a steep weighting curve. Larger $\tau$ values (e.g., $\tau = 2, 5$) flatten the curve, assigning relatively higher weights to hard tokens. The case $\tau = 1$ corresponds to the identity mapping. This illustrates how $\tau$ modulates the balance between emphasizing easy versus hard tokens.

poor results. For example, on Qwen3-4B fine-tuned with MedCalc at a learning rate of $5 \times 10^{-6}$, fixing $\tau = 1$ yields a maximum accuracy of only $0.2168$, much lower than results in Figure 1. This stark degradation confirms that without dynamic adjustment, the model fails to effectively learn from hard tokens. By contrast, our dynamic strategy for selecting $\tau$, i.e., based on the median of average sequence losses in each batch, automatically adapts to the current distribution of token difficulties, ensuring that hard tokens are downweighted without being entirely neglected.

# E  ADDITIONAL DEFINITIONS, THEOREMS AND PROOF

## E.1  PROOF OF THEOREM B.1

---

**Theorem E.1** (First-order approximation by exponential tilting). *Fix a prefix $u$. Consider the current model distribution $q_t(\cdot \mid u)$ and the smoothed target distribution $\tilde{p}_{2,t}(\cdot \mid u)$. Define the local $L^2$ norm $\|g\|_{t,u} := \left(\mathbb{E}_{q_t(\cdot|u)}[g(a)^2]\right)^{1/2}$. Under the standing assumptions, there exists an effective step size $\lambda_{t,u}$, such that*

$$\left\| \log q_{t+1}(\cdot \mid u) - \left[ (1 - \lambda_{t,u}) \log q_t(\cdot \mid u) + \lambda_{t,u} \log \tilde{p}_{2,t}(\cdot \mid u) - \psi_{t,u} \right] \right\|_{t,u} = O(\varepsilon).$$

*where $\psi_{t,u}$ is the log-normalizer and $\varepsilon$ is the KL trust-region radius such that $\mathrm{KL}\big(q_t \,\|\, q_{t+1}\big) \le \varepsilon$.*

---

*Proof.* Fix a prefix $u$. For clarity we write $q_t(\cdot) = q_t(\cdot \mid u)$, $q_{t+1}(\cdot) = q_{t+1}(\cdot \mid u)$, and $\tilde{p}_{2,t}(\cdot) = \tilde{p}_{2,t}(\cdot \mid u)$.

**Step 1. Log-shift representation of the true update.** Define the centered log-shift

$$s(a) \; := \; \log \frac{q_{t+1}(a)}{q_t(a)} - \mathbb{E}_{q_t}\left[ \log \frac{q_{t+1}}{q_t} \right].$$

Then, we have

$$q_{t+1}(a) \; = \; \frac{q_t(a)\, e^{s(a)}}{\mathbb{E}_{q_t}[e^s]}. \tag{2}$$

**Step 2. Log-shift representation of exponential tilting.** For any $\lambda \in [0,1]$, define

$$\widehat{q}^{(\lambda)}(a) \; = \; \frac{q_t(a)\, \exp\{\lambda r(a)\}}{\mathbb{E}_{q_t}[e^{\lambda r}]}, \qquad r(a) := \log \frac{\tilde{p}_{2,t}(a)}{q_t(a)}.$$

**Step 3. Size of the true log-shift (forward KL trust region).** Write $q_t(\cdot) = q_t(\cdot \mid u)$, $q_{t+1}(\cdot) = q_{t+1}(\cdot \mid u)$. Recall the centered log-shift

$$s(a) \; := \; \log \frac{q_{t+1}(a)}{q_t(a)} - \mathbb{E}_{q_t}\left[ \log \frac{q_{t+1}}{q_t} \right], \qquad \|s\|_{t,u}^2 \; := \; \mathbb{E}_{q_t}[s^2].$$

Let the log-partition $A(f) := \log \mathbb{E}_{q_t}[e^f]$ and the exponential-family map

$$T(f)(a) \; := \; \frac{q_t(a)e^{f(a)}}{e^{A(f)}}.$$

Then $q_{t+1} = T(s)$ and

$$\mathrm{KL}\big(q_t \,\|\, q_{t+1}\big) = \mathbb{E}_{q_t}\left[ \log \frac{q_t}{q_{t+1}} \right] = A(s). \tag{3}$$

Two standard identities (for discrete finite support) are

$$\nabla A(f)[h] = \mathbb{E}_{T(f)}[h], \qquad \nabla^2 A(f)[h,k] = \mathrm{Cov}_{T(f)}(h,k).$$

Since $s$ is centered under $q_t$ we have $A(0) = 0$ and $\nabla A(0)[s] = \mathbb{E}_{q_t}[s] = 0$. A second-order Taylor expansion of $A$ at $0$ with a third-order remainder yields

$$A(s) = A(0) + \nabla A(0)[s] + \tfrac{1}{2} \nabla^2 A(0)[s,s] + R_3(s) = \tfrac{1}{2} \mathrm{Var}_{q_t}(s) + R_3(s), \tag{4}$$

where, because the vocabulary is finite and $q_t$ has full support (by smoothing), there exists a constant $C_3 < \infty$ such that

$$|R_3(s)| \; \le \; C_3 \|s\|_{t,u}^3. \tag{5}$$

Combining equation 3–equation 5 gives the quadratic expansion

$$\mathrm{KL}\big(q_t \,\|\, q_{t+1}\big) = \tfrac{1}{2} \|s\|_{t,u}^2 + O\big(\|s\|_{t,u}^3\big).$$

Finally, under the trust-region assumption $\text{KL}(q_t \| q_{t+1}) \leq \varepsilon$ and for $\|s\|_{t,u}$ sufficiently small, there exists a constant $C > 0$ such that

$$\tfrac{1}{2} \|s\|_{t,u}^2 - C \|s\|_{t,u}^3 \;\leq\; \varepsilon,$$

which implies $\|s\|_{t,u} \leq 4\sqrt{\varepsilon}$ as soon as $\|s\|_{t,u} \leq \min\{1/(4C), 1\}$. Hence $\|s\|_{t,u} = O(\sqrt{\varepsilon})$.

**Step 4. First-order expansion of tilting.** Recall the tilted distribution

$$\widehat{q}^{(\lambda)}(a) = \frac{q_t(a)\, e^{\lambda r(a)}}{\mathbb{E}_{q_t}[e^{\lambda r}]}, \qquad r(a) = \log \frac{\tilde{p}_{2,t}(a \mid u)}{q_t(a \mid u)}.$$

Its log-ratio relative to $q_t$ is

$$\log \frac{\widehat{q}^{(\lambda)}(a)}{q_t(a)} = \lambda\, r(a) - A(\lambda r), \qquad A(f) := \log \mathbb{E}_{q_t}[e^f].$$

By Taylor expansion of $A(\lambda r)$ at $\lambda = 0$, using $\nabla A(0)[r] = \mathbb{E}_{q_t}[r]$ and $\nabla^2 A(0)[r, r] = \text{Var}_{q_t}(r)$, one obtains

$$A(\lambda r) = \lambda\, \mathbb{E}_{q_t}[r] + \tfrac{1}{2} \lambda^2\, \text{Var}_{q_t}(r) + \tfrac{1}{6} \lambda^3\, \kappa_3(r) + O(\lambda^4),$$

where $\kappa_3(r)$ denotes the third central moment of $r$ (bounded on finite support). Hence

$$\log \frac{\widehat{q}^{(\lambda)}(a)}{q_t(a)} = \lambda\big(r(a) - \mathbb{E}_{q_t}[r]\big) - \tfrac{1}{2} \lambda^2\, \text{Var}_{q_t}(r) + O(\lambda^3).$$

Now consider the centered log-shift

$$\tilde{s}^{(\lambda)}(a) := \log \frac{\widehat{q}^{(\lambda)}(a)}{q_t(a)} - \mathbb{E}_{q_t}\!\left[\log \frac{\widehat{q}^{(\lambda)}}{q_t}\right].$$

Since

$$\mathbb{E}_{q_t}\!\left[\log \frac{\widehat{q}^{(\lambda)}}{q_t}\right] = \lambda \mathbb{E}_{q_t}[r] - A(\lambda r) = -\tfrac{1}{2} \lambda^2\, \text{Var}_{q_t}(r) + O(\lambda^3),$$

the quadratic terms cancel, yielding

$$\tilde{s}^{(\lambda)}(a) = \lambda\big(r(a) - \mathbb{E}_{q_t}[r]\big) + O(\lambda^3).$$

**Step 5. Choice of effective step size.** Let $r_c(a) := r(a) - \mathbb{E}_{q_t}[r]$ be the centered tilting direction and recall from Step 4 that the centered log-shift of the tilted model satisfies

$$\tilde{s}^{(\lambda)}(a) = \lambda\, r_c(a) + O(\lambda^3).$$

Define the effective step size by $q_t$-least-squares matching:

$$\lambda_{t,u} := \frac{\mathbb{E}_{q_t}\big[s\, r_c\big]}{\mathbb{E}_{q_t}\big[r_c^2\big]}.$$

By Cauchy–Schwarz,

$$|\lambda_{t,u}| = \frac{|\mathbb{E}_{q_t}[s\, r_c]|}{\mathbb{E}_{q_t}[r_c^2]} \leq \frac{\|s\|_{t,u}\, \|r_c\|_{t,u}}{\|r_c\|_{t,u}^2} = \frac{\|s\|_{t,u}}{\|r_c\|_{t,u}}.$$

Since Step 3 gives $\|s\|_{t,u} = O(\sqrt{\varepsilon})$ and we assume $\|r_c\|_{t,u}^2 = \mathbb{E}_{q_t}[r_c^2] \geq v_0 > 0$ (non-degenerate target), it follows that

$$|\lambda_{t,u}| = O(\sqrt{\varepsilon}).$$

Next we control the residual. Under the smoothness and small-step assumptions, the true shift $s$ and the tilting direction $r_c$ agree to first order: there exists a scalar $\alpha = O(\sqrt{\varepsilon})$ and a remainder $\Delta$ with $\|\Delta\|_{t,u} = O(\varepsilon)$ such that

$$s = \alpha\, r_c + \Delta.$$

Substituting this into the formula for $\lambda_{t,u}$ yields

$$\lambda_{t,u} = \frac{\mathbb{E}_{q_t}[(\alpha r_c + \Delta)r_c]}{\mathbb{E}_{q_t}[r_c^2]} = \alpha + \frac{\mathbb{E}_{q_t}[\Delta\, r_c]}{\mathbb{E}_{q_t}[r_c^2]} = \alpha + O(\varepsilon).$$

Hence the residual can be written as

$$s - \lambda_{t,u} r_c = \Delta - (\lambda_{t,u} - \alpha)r_c,$$

and therefore

$$\| s - \lambda_{t,u} r_c \|_{t,u} \le \|\Delta\|_{t,u} + |\lambda_{t,u} - \alpha| \, \|r_c\|_{t,u} = O(\varepsilon).$$

**Step 6. Putting pieces together.** Recall that

$$\log \widehat{q}^{(\lambda)}(a) = (1 - \lambda) \log q_t(a) + \lambda \log \tilde{p}_{2,t}(a) - \psi_{t,u}(\lambda),$$

so that the log-difference vector is

$$\Delta^{(\lambda)}(a) := \log q_{t+1}(a) - \log \widehat{q}^{(\lambda)}(a) = \left( \log \tfrac{q_{t+1}(a)}{q_t(a)} - \mathbb{E}_{q_t}\big[ \log \tfrac{q_{t+1}}{q_t} \big] \right) - \left( \log \tfrac{\widehat{q}^{(\lambda)}(a)}{q_t(a)} - \mathbb{E}_{q_t}\big[ \log \tfrac{\widehat{q}^{(\lambda)}}{q_t} \big] \right) + C(\lambda),$$

where

$$C(\lambda) := \mathbb{E}_{q_t}\Big[ \log \tfrac{q_{t+1}}{q_t} \Big] - \mathbb{E}_{q_t}\Big[ \log \tfrac{\widehat{q}^{(\lambda)}}{q_t} \Big]$$

is a constant (independent of $a$). Denote

$$s(a) := \log \tfrac{q_{t+1}(a)}{q_t(a)} - \mathbb{E}_{q_t}\big[ \log \tfrac{q_{t+1}}{q_t} \big], \qquad \tilde{s}^{(\lambda)}(a) := \log \tfrac{\widehat{q}^{(\lambda)}(a)}{q_t(a)} - \mathbb{E}_{q_t}\big[ \log \tfrac{\widehat{q}^{(\lambda)}}{q_t} \big].$$

Then

$$\Delta^{(\lambda)}(a) = \big(s - \tilde{s}^{(\lambda)}\big)(a) + C(\lambda).$$

Since $\mathbb{E}_{q_t}[s] = \mathbb{E}_{q_t}[\tilde{s}^{(\lambda)}] = 0$, the vector $s - \tilde{s}^{(\lambda)}$ is orthogonal (in $L^2(q_t)$) to the constant function 1. Hence

$$\|\Delta^{(\lambda)}\|_{t,u}^2 = \| s - \tilde{s}^{(\lambda)} \|_{t,u}^2 + |C(\lambda)|^2,$$

and in particular

$$\| s - \tilde{s}^{(\lambda)} \|_{t,u} \le \|\Delta^{(\lambda)}\|_{t,u} \le \| s - \tilde{s}^{(\lambda)} \|_{t,u} + |C(\lambda)|.$$

Recall $A(f) := \log \mathbb{E}_{q_t}[e^f]$ and Eq. 2. Using

$$\mathbb{E}_{q_t}\Big[ \log \tfrac{q_{t+1}}{q_t} \Big] = -A(s), \qquad \mathbb{E}_{q_t}\Big[ \log \tfrac{\widehat{q}^{(\lambda)}}{q_t} \Big] = \lambda \mathbb{E}_{q_t}[r] - A(\lambda r),$$

we have

$$C(\lambda) = -A(s) - \lambda \mathbb{E}_{q_t}[r] + A(\lambda r).$$

By Step 3, $A(s) = \tfrac{1}{2}\|s\|_{t,u}^2 + O(\|s\|_{t,u}^3) = O(\varepsilon)$. By the Taylor expansion of $A(\lambda r)$ at $\lambda = 0$ (Step 4),

$$A(\lambda r) = \lambda \mathbb{E}_{q_t}[r] + \tfrac{1}{2}\lambda^2 \operatorname{Var}_{q_t}(r) + O(\lambda^3).$$

Hence

$$C(\lambda) = -\tfrac{1}{2}\|s\|_{t,u}^2 + \tfrac{1}{2}\lambda^2 \operatorname{Var}_{q_t}(r) + O(\|s\|_{t,u}^3) + O(\lambda^3).$$

In particular, with $|\lambda| = O(\sqrt{\varepsilon})$ (Step 5) and $\|s\|_{t,u} = O(\sqrt{\varepsilon})$ (Step 3),

$$|C(\lambda)| = O(\varepsilon). \tag{6}$$

Choose $\lambda = \lambda_{t,u}$ from Step 5. Then

$$\| s - \tilde{s}^{(\lambda_{t,u})} \|_{t,u} \le \| s - \lambda_{t,u} r_c \|_{t,u} + \|\tilde{s}^{(\lambda_{t,u})} - \lambda_{t,u} r_c\|_{t,u}.$$

By Step 5, $\| s - \lambda_{t,u} r_c \|_{t,u} = O(\varepsilon)$. By Step 4, $\tilde{s}^{(\lambda)} = \lambda r_c + O(\lambda^3)$, so $\|\tilde{s}^{(\lambda_{t,u})} - \lambda_{t,u} r_c\|_{t,u} = O(\lambda_{t,u}^3) = O(\varepsilon^{3/2})$. Therefore,

$$\| s - \tilde{s}^{(\lambda_{t,u})} \|_{t,u} = O(\varepsilon). \tag{7}$$

Using the decomposition inequality above and equation 6,

$$\|\Delta^{(\lambda_{t,u})}\|_{t,u} \leq \| \, s - \tilde{s}^{(\lambda_{t,u})} \, \|_{t,u} + |C(\lambda_{t,u})| = O(\varepsilon) + O(\varepsilon) = O(\varepsilon).$$

Finally, recalling

$$\log \widehat{q}^{(\lambda)}(a) = (1 - \lambda)\log q_t(a) + \lambda \log \tilde{p}_{2,t}(a) - \psi_{t,u}(\lambda),$$

we have shown

$$\left\| \log q_{t+1}(\cdot \mid u) - \Big[(1 - \lambda_{t,u})\log q_t(\cdot \mid u) + \lambda_{t,u} \log \tilde{p}_{2,t}(\cdot \mid u) - \psi_{t,u}(\lambda_{t,u})\Big] \right\|_{t,u} = O(\varepsilon),$$

which proves the theorem.

$\square$

### E.2    PROOF OF THEOREM B.2

**Notation.** Fix a prefix $u$ (we omit "$\mid u$" when clear). Write $f(a) = \log \tilde{p}_2(a) - \log q(a)$ at the current iterate $q$ (the step index $t$ is omitted for readability), and $\bar{f} = \mathbb{E}_q[f]$, $\tilde{f} = f - \bar{f}$. For a set $\mathcal{S}$ of token-tree nodes, denote its $q$-mass by $w_{\mathcal{S}} = \mathbb{E}_q[\mathbf{1}_{\mathcal{S}}]$.

Recall the standard log-space interpolation $Q_\lambda$ defined per prefix by $\log q_\lambda = (1 - \lambda)\log q + \lambda \log \tilde{p}_2 - \psi(\lambda)$, with $\psi(\lambda) = \log \sum_a q(a)^{1-\lambda}\tilde{p}_2(a)^\lambda$.

**Lemma E.1** (First-order change of code length under tilting)**.** *For any response distribution $P$ on the token tree,*

$$\Delta L(P) := \mathrm{KL}(P\|Q_\lambda) - \mathrm{KL}(P\|Q) = -\lambda\Big(\mathbb{E}_P[f] - \mathbb{E}_Q[f]\Big) + O(\lambda^2),$$

*where the $O(\lambda^2)$ remainder is controlled by $\mathrm{Var}_Q(f)$. Equivalently, $\frac{d}{d\lambda}\big|_{\lambda=0}\mathrm{KL}(P\|Q_\lambda) = -\big(\mathbb{E}_P[f] - \mathbb{E}_Q[f]\big)$.*

*Proof.* By definition, $\log q_\lambda = \log q + \lambda(f - \psi(\lambda))$ with $\psi(\lambda) = \log \mathbb{E}_Q[e^{\lambda f}]$. Thus $\log \frac{q_\lambda}{q} = \lambda f - \psi(\lambda)$ and

$$\mathrm{KL}(P\|Q_\lambda) = \mathrm{KL}(P\|Q) - \lambda \mathbb{E}_P[f] + \psi(\lambda).$$

Since $\psi(\lambda) = \log \mathbb{E}_Q[e^{\lambda f}] = \lambda\mathbb{E}_Q[f] + \frac{\lambda^2}{2}\mathrm{Var}_Q(f) + O(\lambda^3)$, we obtain $\Delta L(P) = -\lambda(\mathbb{E}_P[f] - \mathbb{E}_Q[f]) + \frac{\lambda^2}{2}\mathrm{Var}_Q(f) + O(\lambda^3)$. $\square$

**Lemma E.2** (Variance under sparsity)**.** *Fix a prefix $u$ and write $Q(\cdot) = q(\cdot \mid u)$. Let $f(a) = \log \tilde{p}_2(a \mid u) - \log q(a \mid u)$. Under Assumption B.4, we have*

$$\mathrm{Var}_Q(f) \leq \mathbb{E}_{a \sim Q}[f(a)^2] \leq w_{\mathcal{S}} M_h^2 + (1 - w_{\mathcal{S}}) M_l^2 \leq w_{\mathcal{S}} M_h^2 + M_l^2.$$

*Moreover, when we account for the controlled leakage in Assumption B.5, the* effective *out-of-set amplitude can be taken as $M_l + (\beta + \gamma)M_h$, which yields the coarse bound*

$$\mathrm{Var}_Q^{\mathrm{eff}}(f) \leq w_{\mathcal{S}} M_h^2 + \big(M_l + (\beta + \gamma)M_h\big)^2.$$

*Proof.* Since $\mathrm{Var}_Q(f) \leq \mathbb{E}_Q[f^2]$, it suffices to bound the second moment. Split the expectation over $\mathcal{S}$ and $\mathcal{S}^c$:

$$\mathbb{E}_Q[f^2] = \mathbb{E}_Q[f^2 \mathbf{1}_{\mathcal{S}}] + \mathbb{E}_Q[f^2 \mathbf{1}_{\mathcal{S}^c}] \leq w_{\mathcal{S}} M_h^2 + (1 - w_{\mathcal{S}})M_l^2,$$

using $|f| \leq M_h$ on $\mathcal{S}$ and $|f| \leq M_l$ on $\mathcal{S}^c$. Dropping the factor $(1 - w_{\mathcal{S}})$ gives $\leq w_{\mathcal{S}} M_h^2 + M_l^2$.

To upper bound the *effective* out-of-set magnitude after realizing a targeted update on $\mathcal{S}$, choose the per-prefix target direction

$$g := \Pi_{\mathcal{S}} f,$$

which is supported on $\mathcal{S}$, so that $\|\Pi_{\mathcal{S}^c} g\|_{L^2(Q)} = 0 \leq \beta \, \|\Pi_{\mathcal{S}} g\|_{L^2(Q)}$ and the premise of Assumption B.5 holds. Let $\Delta \log q$ be the induced global change guaranteed by Assumption B.5, and define the *leakage vector* on $\mathcal{S}^c$ by

$$\ell := \Pi_{\mathcal{S}^c}\,\Delta \log q.$$

Then the out-of-set control in Assumption B.5 gives

$$\|\ell\|_{L^2(Q)} \le (\beta + \gamma) \, \|\Pi_{\mathcal{S}} g\|_{L^2(Q)}.$$

Moreover,

$$\|\Pi_{\mathcal{S}} g\|_{L^2(Q)} = \|\Pi_{\mathcal{S}} f\|_{L^2(Q)} = \left( \mathbb{E}_{a \sim Q} \big[ (\Pi_{\mathcal{S}} f(a))^2 \big] \right)^{1/2} \le M_h \, \sqrt{w_{\mathcal{S}}} \le M_h.$$

Define the *effective* out-of-set component that accounts for leakage by

$$f_{\mathcal{S}^c}^{\mathrm{eff}} := f_{\mathcal{S}^c} + \ell.$$

By the triangle inequality and the bounds above,

$$\|f_{\mathcal{S}^c}^{\mathrm{eff}}\|_{L^2(Q)} \le \|f_{\mathcal{S}^c}\|_{L^2(Q)} + \|\ell\|_{L^2(Q)} \le M_l + (\beta + \gamma) \, \|\Pi_{\mathcal{S}} g\|_{L^2(Q)} \le M_l + (\beta + \gamma) \, M_h.$$

Therefore, an effective second-moment upper bound that incorporates the controlled leakage is

$$\mathbb{E}_Q[f^2] = \|f_{\mathcal{S}}\|_{L^2(Q)}^2 + \|f_{\mathcal{S}^c}\|_{L^2(Q)}^2 \le w_{\mathcal{S}} \, M_h^2 + \big( M_l + (\beta + \gamma) M_h \big)^2.$$

This motivates the shorthand

$$\mathrm{Var}_Q^{\mathrm{eff}}(f) \; \le \; w_{\mathcal{S}} \, M_h^2 + \big( M_l + (\beta + \gamma) M_h \big)^2,$$

which we use as a coarse variance upper bound when controlled leakage is present. $\qquad \square$

**Lemma E.3.** *Let $P, Q, \tilde{P}_2$ be response distributions on the (truncated) token space with strictly positive densities $p, q, \tilde{p}_2$. Define $f(z) := \log \tilde{p}_2(z) - \log q(z)$. Then*

$$\mathbb{E}_P[f] - \mathbb{E}_Q[f] = \mathrm{KL}(P\|Q) + \mathrm{KL}(Q\|\tilde{P}_2) - \mathrm{KL}(P\|\tilde{P}_2).$$

*In particular, if $\tilde{P}_2 = P_2$ then $\mathbb{E}_{P_2}[f] - \mathbb{E}_Q[f] = \mathrm{KL}(P_2\|Q) + \mathrm{KL}(Q\|P_2) \ge 0$.*

*Proof.* Expand the left-hand side:

$$\sum_z p(z) \big( \log \tilde{p}_2(z) - \log q(z) \big) - \sum_z q(z) \big( \log \tilde{p}_2(z) - \log q(z) \big).$$

Group like terms:

$$\Big( \sum_z p(z) \log \tilde{p}_2(z) - \sum_z p(z) \log q(z) \Big) - \Big( \sum_z q(z) \log \tilde{p}_2(z) - \sum_z q(z) \log q(z) \Big).$$

Use the discrete KL definitions:

$$\mathrm{KL}(P\|Q) = \sum_z p(z) \log \frac{p(z)}{q(z)} = \sum_z p(z) \log p(z) - \sum_z p(z) \log q(z),$$

$$\mathrm{KL}(Q\|\tilde{P}_2) = \sum_z q(z) \log \frac{q(z)}{\tilde{p}_2(z)} = \sum_z q(z) \log q(z) - \sum_z q(z) \log \tilde{p}_2(z),$$

$$\mathrm{KL}(P\|\tilde{P}_2) = \sum_z p(z) \log \frac{p(z)}{\tilde{p}_2(z)} = \sum_z p(z) \log p(z) - \sum_z p(z) \log \tilde{p}_2(z).$$

Substitute and simplify to obtain

$$\sum_z p(z) f(z) - \sum_z q(z) f(z) = \mathrm{KL}(P\|Q) + \mathrm{KL}(Q\|\tilde{P}_2) - \mathrm{KL}(P\|\tilde{P}_2).$$

For $\tilde{P}_2 = P_2$, the right-hand side equals $\mathrm{KL}(P_2\|Q) + \mathrm{KL}(Q\|P_2) \ge 0$, with equality iff $Q = P_2$. $\qquad \square$

We now bound the domain performance improvement and the general capability degradation in a single small step.

---

**Theorem E.2** (One-step code-length change bounds). *Let $P_1, P_2$ be the response distributions of $\mathcal{D}_1, \mathcal{D}_2$ on the truncated token tree, and let $Q \mapsto Q_\lambda$ be one log-space tilting step with $|\lambda| \leq \lambda_0$ small. Denote the expected code-length change by $\Delta L(P) := \mathrm{KL}(P\|Q_\lambda) - \mathrm{KL}(P\|Q)$. Then there exist constants $C_1, C_2 \geq 0$ such that:*

*Domain performance improvement on $\mathcal{D}_2$.*

$$\Delta L(P_2) = -\lambda\Big(\mathbb{E}_{P_2}[f] - \mathbb{E}_Q[f]\Big) + \tfrac{\lambda^2}{2}\mathrm{Var}_Q(f) + O(\lambda^3)$$

$$\leq -\lambda\Big(\mathrm{KL}(Q\|\tilde{P}_2) - \mathrm{KL}(P_2\|\tilde{P}_2)\Big) + C_2\,\lambda^2.$$

*General performance degradation on $\mathcal{D}_1$.*

$$\Delta L(P_1) = -\lambda\Big(\mathbb{E}_{P_1}[f] - \mathbb{E}_Q[f]\Big) + \tfrac{\lambda^2}{2}\mathrm{Var}_Q(f) + O(\lambda^3)$$

$$\leq \lambda\sqrt{\mathrm{Var}_Q(f)}\sqrt{\chi^2(P_1\|Q)} + C_1\,\lambda^2$$

$$\leq \lambda\Big(\sqrt{w_{\mathcal{S}}}\,M_h + M_l + (\beta + \gamma)M_h\Big)\sqrt{\chi^2(P_1\|Q)} + C_1\,\lambda^2,$$

*where $\chi^2(P_1\|Q)$ is the chi-square divergence.*

*In particular, if $\tilde{P}_2 = P_2$, then*

$$\Delta L(P_2) \leq -\lambda\,\mathrm{KL}(Q\|P_2) + C_2\,\lambda^2.$$

---

*Proof.* Recall the definition of the one-step log-space interpolation (per prefix) $\log q_\lambda = (1 - \lambda)\log q + \lambda\log\tilde{p}_2 - \psi(\lambda)$, with $\psi(\lambda) = \log\mathbb{E}_Q[e^{\lambda f}]$ and $f = \log\tilde{p}_2 - \log q$. For any distribution $P$ on responses, Lemma E.1 gives the second-order expansion

$$\Delta L(P) := \mathrm{KL}(P\|Q_\lambda) - \mathrm{KL}(P\|Q) = -\lambda\big(\mathbb{E}_P[f] - \mathbb{E}_Q[f]\big) + \tfrac{\lambda^2}{2}\mathrm{Var}_Q(f) + O(\lambda^3). \quad (8)$$

**Domain performance improvement on $\mathcal{D}_2$.** Apply equation 8 with $P = P_2$:

$$\Delta L(P_2) = -\lambda\big(\mathbb{E}_{P_2}[f] - \mathbb{E}_Q[f]\big) + \tfrac{\lambda^2}{2}\mathrm{Var}_Q(f) + O(\lambda^3).$$

By Lemma E.3,

$$\mathbb{E}_{P_2}[f] - \mathbb{E}_Q[f] = \mathrm{KL}(P_2\|Q) + \mathrm{KL}(Q\|\tilde{P}_2) - \mathrm{KL}(P_2\|\tilde{P}_2) \geq \mathrm{KL}(Q\|\tilde{P}_2) - \mathrm{KL}(P_2\|\tilde{P}_2).$$

Multiplying by $-\lambda$ reverses the inequality, hence for $|\lambda| \leq \lambda_0$,

$$\Delta L(P_2) \leq -\lambda\Big(\mathrm{KL}(Q\|\tilde{P}_2) - \mathrm{KL}(P_2\|\tilde{P}_2)\Big) + C_2\,\lambda^2,$$

where $C_2 \geq 0$. In particular, when $\tilde{P}_2 = P_2$,

$$\mathbb{E}_{P_2}[f] - \mathbb{E}_Q[f] = \mathrm{KL}(P_2\|Q) + \mathrm{KL}(Q\|P_2) \geq \mathrm{KL}(Q\|P_2),$$

so

$$\Delta L(P_2) \leq -\lambda\,\mathrm{KL}(Q\|P_2) + C_2\,\lambda^2.$$

**General performance degradation on $\mathcal{D}_1$.** Apply equation 8 with $P = P_1$:

$$\Delta L(P_1) = -\lambda\big(\mathbb{E}_{P_1}[f] - \mathbb{E}_Q[f]\big) + \tfrac{\lambda^2}{2}\mathrm{Var}_Q(f) + O(\lambda^3).$$

Using the inequality $|\mathbb{E}_P[g] - \mathbb{E}_Q[g]| \leq \sqrt{\mathrm{Var}_Q(g)}\sqrt{\chi^2(P\|Q)}$, we have:

$$\big|\mathbb{E}_{P_1}[f] - \mathbb{E}_Q[f]\big| \leq \sqrt{\chi^2(P_1\|Q)}\sqrt{\mathrm{Var}_Q(f)}.$$

Therefore, for $|\lambda| \leq \lambda_0$,

$$\Delta L(P_1) \leq \lambda \sqrt{\mathrm{Var}_Q(f)} \sqrt{\chi^2(P_1 \| Q)} + C_1 \lambda^2,$$

with $C_1 \geq 0$ as above.

Next, invoke the variance upper bound that incorporates the controlled leakage (Lemma E.2):

$$\mathrm{Var}_Q(f) \leq \mathrm{Var}_Q^{\mathrm{eff}}(f) \leq w_{\mathcal{S}} M_h^2 + \left(M_l + (\beta + \gamma)M_h\right)^2.$$

Taking square-roots and using $\sqrt{x+y} \leq \sqrt{x} + \sqrt{y}$,

$$\sqrt{\mathrm{Var}_Q(f)} \leq \sqrt{\mathrm{Var}_Q^{\mathrm{eff}}(f)} \leq \sqrt{w_{\mathcal{S}}}\, M_h + M_l + (\beta + \gamma)M_h.$$

Combine the last two displays to obtain

$$\Delta L(P_1) \leq \lambda \left(\sqrt{w_{\mathcal{S}}}\, M_h + M_l + (\beta + \gamma)M_h\right)\sqrt{\chi^2(P_1 \| Q)} + C_1 \lambda^2.$$

This establishes both bounds with the left-hand side written as the expected code-length change $\Delta L(P)$. $\qquad\square$

---

**Theorem E.3** (Multi-step code-length change bounds). *Let $(Q_t)_{t=0}^T$ be obtained by repeated log-space tilting steps $Q_t \mapsto Q_{t+1}$ with weights $\lambda_t$ (per-prefix interpolation), and write*

$$\Delta L_T(P) := \mathrm{KL}(P\|Q_T) - \mathrm{KL}(P\|Q_0), \qquad \Lambda_T := \sum_{t=0}^{T-1} \lambda_t, \qquad S_T := \sum_{t=0}^{T-1} \lambda_t^2.$$

*Then there exist constants $C_1, C_2 \geq 0$ such that:*

*Domain performance improvement on $\mathcal{D}_2$.*

$$\Delta L_T(P_2) = \sum_{t=0}^{T-1} \left(\mathrm{KL}(P_2\|Q_{t+1}) - \mathrm{KL}(P_2\|Q_t)\right) \leq -\sum_{t=0}^{T-1} \lambda_t \left(\mathbb{E}_{P_2}[f_t] - \mathbb{E}_{Q_t}[f_t]\right) + C_2\, S_T.$$

*In particular, if $\tilde{P}_2 = P_2$ (oracle targets), then*

$$\Delta L_T(P_2) \leq -\sum_{t=0}^{T-1} \lambda_t\, \mathrm{KL}(Q_t\|P_2) + C_2\, S_T.$$

*General performance degradation on $\mathcal{D}_1$. Let*

$$H_T := \sup_{0 \leq t < T} \sqrt{\chi^2(P_1\|Q_t)} \quad (< \infty \text{ under small-step updates and } \mathrm{KL}(P_1\|Q_0) \ll 1).$$

*Then*

$$\Delta L_T(P_1) \leq H_T \sum_{t=0}^{T-1} \lambda_t \sqrt{\mathrm{Var}_{Q_t}(f_t)} + C_1\, S_T \leq H_T\left(\sqrt{w_{\mathcal{S}}}\, M_h + M_l + (\beta + \gamma)M_h\right)\Lambda_T + C_1\, S_T$$

*where the last inequality uses the effective variance bound from Lemma E.2, applied uniformly over $t$.*

---

*Proof.* For each step $t$, Lemma E.1 gives, for any $P$,

$$\mathrm{KL}(P\|Q_{t+1}) - \mathrm{KL}(P\|Q_t) = -\lambda_t\left(\mathbb{E}_P[f_t] - \mathbb{E}_{Q_t}[f_t]\right) + \frac{\lambda_t^2}{2}\mathrm{Var}_{Q_t}(f_t) + O(\lambda_t^3).$$

Summing over $t = 0, \dots, T-1$ and absorbing $\sum_t O(\lambda_t^3)$ into a constant multiple of $S_T$ (since $|\lambda_t| \leq \lambda_0$) yields the generic decomposition

$$\Delta L_T(P) \leq -\sum_{t=0}^{T-1} \lambda_t\left(\mathbb{E}_P[f_t] - \mathbb{E}_{Q_t}[f_t]\right) + C\, S_T.$$

*Domain term ($P = P_2$).* By Lemma E.3, $\mathbb{E}_{P_2}[f_t] - \mathbb{E}_{Q_t}[f_t] = \mathrm{KL}(P_2\|Q_t) + \mathrm{KL}(Q_t\|\tilde{P}_2) - \mathrm{KL}(P_2\|\tilde{P}_2) \geq \mathrm{KL}(Q_t\|\tilde{P}_2) - \mathrm{KL}(P_2\|\tilde{P}_2)$. Multiplying by $-\lambda_t$ and summing gives the first display, with $C_2$ absorbing the quadratic/cubic remainders. In the oracle case $\tilde{P}_2 = P_2$, $\mathbb{E}_{P_2}[f_t] - \mathbb{E}_{Q_t}[f_t] \geq \mathrm{KL}(Q_t\|P_2)$, hence the stated inequality.

*General term ($P = P_1$).* The change-of-measure bound with centering gives, for each $t$,

$$\left| \mathbb{E}_{P_1}[f_t] - \mathbb{E}_{Q_t}[f_t] \right| \ \leq \ \sqrt{\chi^2(P_1\|Q_t)} \, \sqrt{\mathrm{Var}_{Q_t}(f_t)} \ \leq \ H_T \sqrt{\mathrm{Var}_{Q_t}(f_t)}.$$

Thus, by applying $-(\mathbb{E}_{P_1}[f_t] - \mathbb{E}_{Q_t}[f_t]) \leq \left| \mathbb{E}_{P_1}[f_t] - \mathbb{E}_{Q_t}[f_t] \right|$, we have

$$\Delta L_T(P_1) \ \leq \ H_T \sum_{t=0}^{T-1} \lambda_t \sqrt{\mathrm{Var}_{Q_t}(f_t)} \ + \ C_1 S_T.$$

Finally, apply Lemma E.2 uniformly in $t$ to bound $\sqrt{\mathrm{Var}_{Q_t}(f_t)} \ \leq \ \sqrt{w_{\mathcal{S}}} \, M_h + M_l + (\beta + \gamma) M_h$ and factor out $\Lambda_T$. $\qquad\square$

---

**Theorem E.4** (Smaller steps yield a smaller general performance degradaton bound at a equal domain performance gain). *Fix a desired domain improvement $\Delta_\star > 0$ on $\mathcal{D}_2$ (i.e., $\Delta L_T(P_2) \leq -\Delta_\star$). Among all $T$-step tilting schedules that achieve this target, the* minimal *upper bound on the increase of code length on $\mathcal{D}_1$ satisfies*

$$\Delta L_T(P_1) \ \leq \ A \frac{\Delta_\star}{\mu_T} \ + \ \left( \frac{A C_2}{\mu_T^3} + \frac{C_1}{\mu_T^2} \right) \frac{\Delta_\star^2}{T} \ + \ O\left( \frac{1}{T^2} \right)$$

*where $\mu_T := \inf_{t<T} \mathrm{KL}(Q_t\|P_2) > 0$ and $A := H_T\left( \sqrt{w_{\mathcal{S}}} \, M_h + M_l + (\beta + \gamma) M_h \right)$ are fixed value under the total number of update steps $T$ and the desired domain gain $\Delta_\star$.*
*The upper bound strictly decreases as $T$ increases. Thus, under the equal-steps schedule that attains the target, the per-step effective weight scales as $\lambda_t \propto 1/T$; thus, for the same domain gain, larger $T$ implies smaller per-step updates. Hence, smaller step size $\Rightarrow$ smaller upper bound.*

---

*Proof.* We work in the oracle case $\tilde{P}_2 = P_2$. From the multi-step bound,

$$\Delta L_T(P_2) \ \leq \ - \sum_{t=0}^{T-1} \lambda_t \, \mathrm{KL}(Q_t\|P_2) \ + \ C_2 \sum_{t=0}^{T-1} \lambda_t^2 \ \leq \ -\mu_T \Lambda_T + C_2 S_T,$$

where $\Lambda_T = \sum_t \lambda_t$, $S_T = \sum_t \lambda_t^2$, and $\mu_T := \inf_{t<T} \mathrm{KL}(Q_t\|P_2) > 0$. Thus any schedule that achieves $\Delta L_T(P_2) \leq -\Delta_\star$ must satisfy the feasibility constraint

$$\mu_T \Lambda_T - C_2 S_T \ \geq \ \Delta_\star. \tag{9}$$

For the general-performance side, the multi-step bound gives

$$\Delta L_T(P_1) \ \leq \ A \Lambda_T + C_1 S_T, \tag{10}$$

where $A := H_T\left( \sqrt{w_{\mathcal{S}}} \, M_h + M_l + (\beta + \gamma) M_h \right)$ and $H_T := \sup_{t<T} \sqrt{\chi^2(P_1\|Q_t)}$.

For any fixed $T$ and $\Lambda_T$, Cauchy–Schwarz implies $S_T \geq \Lambda_T^2/T$, with equality iff $\lambda_t \equiv \Lambda_T/T$. Hence the bound equation 10 is minimized (for fixed $T, \Lambda_T$) by the equal-steps schedule; moreover, equal steps minimize the feasibility penalty in equation 9 as well. Under equal steps $S_T = \Lambda_T^2/T$, the feasibility constraint becomes the concave quadratic inequality

$$\mu_T \Lambda_T - \frac{C_2}{T} \Lambda_T^2 \ \geq \ \Delta_\star.$$

Its smallest feasible solution (the smaller root) is

$$\Lambda_T^{\min} \ = \ \frac{T}{2C_2} \left( \mu_T - \sqrt{\mu_T^2 - \frac{4C_2 \Delta_\star}{T}} \right),$$

which exists for $T > 4C_2\Delta_\star/\mu_T^2$. Substituting $\Lambda_T^{\min}$ and $S_T = (\Lambda_T^{\min})^2/T$ into equation 10 yields the optimal-in-this-bound upper bound.

To expose the dependence on $T$, expand the smaller root for large $T$:

$$\mu_T\,\Lambda_T - \frac{C_2}{T}\,\Lambda_T^2 = \Delta_\star,$$

the smaller root is

$$\Lambda_T^{\min} = \frac{T}{2C_2}\left(\mu_T - \sqrt{\mu_T^2 - \frac{4C_2\Delta_\star}{T}}\right).$$

Set

$$\varepsilon := \frac{4C_2\Delta_\star}{T}, \qquad x := \frac{\varepsilon}{\mu_T^2} = \frac{4C_2\Delta_\star}{\mu_T^2\,T}.$$

Then

$$\sqrt{\mu_T^2 - \varepsilon} = \mu_T\sqrt{1 - x} = \mu_T\left(1 - \frac{x}{2} - \frac{x^2}{8} + O(x^3)\right).$$

Hence

$$\mu_T - \sqrt{\mu_T^2 - \varepsilon} = \mu_T\left(\frac{x}{2} + \frac{x^2}{8} + O(x^3)\right) = \frac{\varepsilon}{2\mu_T} + \frac{\varepsilon^2}{8\mu_T^3} + O\left(\frac{\varepsilon^3}{\mu_T^5}\right).$$

Multiplying by the prefactor $T/(2C_2)$ and substituting $\varepsilon = 4C_2\Delta_\star/T$,

Then, we have

$$\Lambda_T^{\min} = \frac{\Delta_\star}{\mu_T} + \frac{C_2\,\Delta_\star^2}{\mu_T^3}\cdot\frac{1}{T} + O\left(\frac{1}{T^2}\right).$$

Therefore,

$$\Delta L_T(P_1) \;\leq\; A\frac{\Delta_\star}{\mu_T} \;+\; \left(\frac{A\,C_2}{\mu_T^3} + \frac{C_1}{\mu_T^2}\right)\frac{\Delta_\star^2}{T} \;+\; O\left(\frac{1}{T^2}\right),$$

which decreases strictly in $T$ and converges to $A\,\Delta_\star/\mu_T$ as $T \to \infty$. This completes the proof. $\square$

---

**Theorem E.5** (Label-only supervision enlarges the safe per-step range). *Define*

$$V_s \;:=\; \sqrt{\mathbb{E}\big[s\,M_h^2 + (m - s)\,M_e^2\big]}, \qquad M_e := M_l + (\beta + \gamma)M_h,$$

*where $s$ is the expected number of hard tokens per example on $\mathcal{D}_2$ and $m$ is the example to-ken length. For any $T$-step equal-steps schedule ($\lambda_t \equiv \lambda$), a general-performance degradation $\Delta L_T(P_1) \leq \varepsilon_{\mathrm{fg}}$ is ensured whenever the per-step effective weight satisfies*

$$\lambda \;\leq\; \lambda_{\max}(T; s) \;:=\; \frac{-H_T V_s + \sqrt{(H_T V_s)^2 + \frac{4C_1}{T}\varepsilon_{\mathrm{fg}}}}{2C_1}.$$

*In particular, as $T$ grows,*

$$\lambda_{\max}(T; s) \;=\; \frac{\varepsilon_{\mathrm{fg}}}{H_T V_s}\cdot\frac{1}{T} \;+\; O\left(\frac{1}{T^2}\right),$$

*so the* safe per-step range *widens inversely with $V_s$. When $M_e \ll M_h$, we have $V_s \asymp M_h\sqrt{s}$, hence*

$$\lambda_{\max}(s) \;\asymp\; \frac{1}{\sqrt{s}} \quad \text{(for fixed $\varepsilon_{\mathrm{fg}}, H_T, M_h, T$)}.$$

*Therefore, if label-only supervision reduces $s$ relative to chain-of-thought, it strictly enlarges the admissible per-step range.*

---

*Proof.* Define $M_e := M_l + (\beta + \gamma)M_h$. From Lemma E.2, we have

$$\mathrm{Var}_{Q_t}(f_t) \;\leq\; \mathbb{E}_{a\sim Q_t}[f_t(a)^2] \;\leq\; w_{\mathcal{S},t}\,M_h^2 \;+\; (1 - w_{\mathcal{S},t})\,M_e^2, \tag{11}$$

Consider sampling an example $z$ from $\mathcal{D}_2$, with response length $m(z) \in \mathbb{N}$, and let $J(z) \subseteq \{1, \ldots, m(z)\}$ be the set of hard positions for that example, with cardinality $s(z) = |J(z)|$. If

we (conceptually) select a token position uniformly at random along the generated path, then the probability of landing in the hard set equals the *hard fraction*:

$$w_{\mathcal{S},t} \;=\; \mathbb{E}\!\left[\frac{s(z)}{m(z)}\right] \;=:\; \mathbb{E}\!\left[\tfrac{s}{m}\right], \tag{12}$$

where the expectation is over the (data-induced) randomness of examples and the path under $Q_t$. Substituting equation 12 into equation 11 yields

$$\mathrm{Var}_{Q_t}(f_t) \;\leq\; \mathbb{E}\!\left[\tfrac{s}{m}\right] M_h^2 \;+\; \left(1 - \mathbb{E}\!\left[\tfrac{s}{m}\right]\right) M_e^2 \,. \tag{13}$$

We now relax the hard fraction into an affine form in $(s, m)$ that pairs naturally with $(M_h^2, M_e^2)$. Since $m(z) \geq 1$ and $0 \leq s(z) \leq m(z)$ for every example,

$$\frac{s(z)}{m(z)} \;\leq\; s(z), \qquad 1 - \frac{s(z)}{m(z)} \;\leq\; m(z) - s(z).$$

Taking expectations and using linearity,

$$\mathbb{E}\!\left[\tfrac{s}{m}\right] M_h^2 \;+\; \left(1 - \mathbb{E}\!\left[\tfrac{s}{m}\right]\right) M_e^2 \;\leq\; \mathbb{E}\!\left[s\, M_h^2 + (m - s)\, M_e^2\right]. \tag{14}$$

Combining equation 13 and equation 14 gives the *uniform (in t)* token-level bound

$$\sqrt{\mathrm{Var}_{Q_t}(f_t)} \;\leq\; \sqrt{\mathbb{E}\!\left[s\, M_h^2 + (m - s)\, M_e^2\right]} \;:=\; V_s, \qquad \text{for all } t = 0, 1, \ldots, T - 1. \tag{15}$$

The general-performance part of Theorem E.3 states that

$$\Delta L_T(P_1) \;\leq\; H_T \sum_{t=0}^{T-1} \lambda_t \sqrt{\mathrm{Var}_{Q_t}(f_t)} \;+\; C_1 \sum_{t=0}^{T-1} \lambda_t^2, \qquad H_T \;:=\; \sup_{0 \leq t < T} \sqrt{\chi^2(P_1 \| Q_t)}.$$

Using equation 15, we obtain the uniform (in $t$) upper bound

$$\Delta L_T(P_1) \;\leq\; H_T\, V_s\, \Lambda_T \;+\; C_1\, S_T, \qquad \Lambda_T := \sum_{t=0}^{T-1} \lambda_t, \;\; S_T := \sum_{t=0}^{T-1} \lambda_t^2. \tag{16}$$

For an equal-steps schedule $\lambda_t \equiv \lambda$, we have $\Lambda_T = T\lambda$ and $S_T = T\lambda^2$, hence from equation 16

$$\Delta L_T(P_1) \;\leq\; T\!\left(H_T\, V_s\, \lambda \;+\; C_1\, \lambda^2\right). \tag{17}$$

Impose a general-performance budget $\Delta L_T(P_1) \leq \varepsilon_{\mathrm{fg}}$. Then equation 17 yields the quadratic constraint

$$C_1\, \lambda^2 \;+\; (H_T V_s)\, \lambda \;-\; \frac{\varepsilon_{\mathrm{fg}}}{T} \;\leq\; 0\,. \tag{18}$$

The feasible interval in $\lambda$ is $[\,0, \; \lambda_{\max}(T; s)\,]$, where the positive root is

$$\lambda_{\max}(T; s) \;=\; \frac{-H_T V_s \;+\; \sqrt{(H_T V_s)^2 \;+\; \frac{4C_1}{T}\, \varepsilon_{\mathrm{fg}}}}{2C_1}\,. \tag{19}$$

A first-order expansion in $1/T$ (Taylor for $\sqrt{a^2 + \delta}$ with $\delta \sim T^{-1}$) gives

$$\lambda_{\max}(T; s) \;=\; \frac{\varepsilon_{\mathrm{fg}}}{H_T V_s} \cdot \frac{1}{T} \;+\; O\!\left(\frac{1}{T^2}\right), \quad \text{as } T \to \infty. \tag{20}$$

Finally, when $M_e \ll M_h$, the $M_e$-term is negligible and $V_s \;=\; \sqrt{\mathbb{E}[s]\, M_h^2 + \mathbb{E}[m - s]\, M_e^2} \;\approx\; M_h \sqrt{s}$, so

$$\lambda_{\max}(T; s) \;\asymp\; \frac{1}{\sqrt{s}} \qquad (\text{for fixed } \varepsilon_{\mathrm{fg}},\, H_T,\, M_h, T).$$

This shows that *reducing s* (as is typical under label-only supervision versus chain-of-thought) *strictly enlarges* the admissible per-step range, and the range also increases with $T$. □

Table 4: General-purpose benchmarks, their evaluation metrics, the number of few-shot settings, and the primary capability they assess.

| Dataset | Metric | Shot | Capability Evaluated |
|---|---|---|---|
| *Instruction Following* | | | |
| IFEval (Zhou et al., 2023) | inst_level_strict_acc, none | 0-shot | Instruction-following |
| *Mathematical Reasoning* | | | |
| GSM8K (Cobbe et al., 2021) | exact_match, flexible-extract | 5-shot | Mathematical reasoning |
| *Code Generation* | | | |
| HumanEval (Chen et al., 2021) | pass@1, create_test | 0-shot | Code generation |
| *Commonsense Reasoning* | | | |
| HellaSwag (Zellers et al., 2019) | acc, none | 0-shot | Commonsense reasoning |
| ARC-Easy (Clark et al., 2018) | acc, none | 0-shot | Science reasoning |
| ARC-Challenge (Clark et al., 2018) | acc, none | 0-shot | Science reasoning |
| PIQA (Bisk et al., 2020) | acc, none | 0-shot | Physical commonsense reasoning |
| *Knowledge-Intensive QA* | | | |
| MMLU (Hendrycks et al., 2020) | acc, none | 0-shot | Multi-domain knowledge understanding |

Table 5: Domain-specific datasets, their evaluation metrics, and the primary capability they assess.

| Dataset | Metric | Domain-Specific Capability Evaluated |
|---|---|---|
| MedCalc (Khandekar et al., 2024) | Accuracy | Medical mathematical reasoning |
| ESCI (Reddy et al., 2022) | Balanced Accuracy | E-commerce product classification |

Table 6: Prompt template used for the MedCalc Benchmark.

**Prompt Template for MedCalc Benchmark.**

```
<|im_start|>system
You are a helpful assistant.  You first think about the
reasoning process in the mind and then provide the user with
the answer.
<|im_end|>

<|im_start|>user
You are a helpful assistant for calculating a score for a
given patient note.  Please think step-by-step to solve the
question and then generate the required score.
Here is the patient note:
{note}

Here is the task:
{question}

Please show your entire reasoning process in a single <think>
</think> block (do not open or close the tag more than once).
Your final response must be in JSON format within <answer>
</answer> tags.  For example,
<think>
[entire reasoning process here]
</think>

<answer>
{"answer":  str(short_and_direct_answer_of_the_question)}
</answer>

<|im_end|>

<|im_start|>assistant
Let me solve this step by step.
<think>
```

Table 7: Prompt template used for the ESCI classification task in the *w/o CoT* setting, where the LLM directly predicts one of four relation types given a query–product pair without generating intermediate reasoning.

**Prompt Template for ESCI Classification (w/o CoT setting)**

```
<|im_start|>system
You are a helpful assistant.  You provide the user with the
answer.
<|im_end|>

<|im_start|>user
Your task is to classify each product as being an Exact,
Substitute, Complement, or Irrelevant match for the query.
Here is the user's query:
{user_query}

Here is the product information:
{product_info}
--------------

Your final response must be within <answer> </answer> tags.
Label the relation type as a number:  0 = Exact, 1 =
Substitute, 2 = Complement and 3 = Irrelevant.  For example,
<answer>one_of_[0, 1, 2, 3]</answer>.
<|im_end|>

<|im_start|>assistant
<answer>
```

Table 8: Prompt template used for the ESCI classification task in the *w/ CoT* setting, where the LLM is required to produce a complete reasoning trace inside a single `<think>` block before giving the final prediction in `<answer>` tags.

---

**Prompt Template for ESCI Classification (w/ CoT setting)**

---

```
<|im_start|>system
You are a helpful assistant.  You first think about the
reasoning process in the mind and then provide the user with
the answer.
<|im_end|>

<|im_start|>user
Your task is to classify each product as being an Exact,
Substitute, Complement, or Irrelevant match for the query.
Here is the user's query:
{user_query}

Here is the product information:
{product_info}
-------------

Please show your entire reasoning process in **a single**
<think> </think> block (do not open or close the tag more than
once).
Your final response must be within <answer> </answer> tags.
Label the relation type as a number:  0 = Exact, 1 =
Substitute, 2 = Complement and 3 = Irrelevant.  For example,
<think>
[entire reasoning process here]
</think>
<answer>one_of_[0, 1, 2, 3]</answer>
<|im_end|>

<|im_start|>assistant
<think>
```

---

