# OpenReview forum: "SFT Doesn’t Always Hurt General Capabilities: Revisiting Domain-Specific Fine-Tuning in LLMs"
_ICLR.cc/2026/Conference — ICLR 2026 Poster_

### Official Review · Reviewer_T4Nt · 2025-10-25

**Soundness:** 3
**Presentation:** 3
**Contribution:** 2
**Rating:** 4
**Confidence:** 4

**Summary:**

The paper's premise is the observation that SFT does not always significantly harm a model's general capabilities ; this issue can, in fact, be largely mitigated by using a smaller learning rate. A theoretical analysis further suggests this performance degradation is linked to the adverse impact of low-probability tokens. Based on this, the authors propose TALR, a method that suppresses the decline in general capabilities by adaptively reducing the loss weights of these "hard" tokens.

Experiments show that TALR outperforms existing methods in balancing domain-specific and general performance , though it cannot fully resolve the trade-off. The paper's final practical guideline for SFT is to prioritize using a small learning rate and to supplement with TALR when a better balance is desired.

**Strengths:**

1. Experiments across multiple models (Qwen and Gemma series) and diverse datasets (MedCalc, ESCI, MetaMathQA) consistently demonstrate the effectiveness of using a small learning rate.
2. The paper moves beyond mere empirical observation to provide a theoretical foundation. The perspective linking SFT dynamics to information compression and code-length change is novel. This theory successfully explains why small learning rates are beneficial and why label-only SFT behaves differently.
3. The proposed TALR method is not ad-hoc; it is derived from the theoretical analysis, giving it a strong theoretical grounding.

**Weaknesses:**

1. The primary recommendation—using a small learning rate—is a standard and well-known heuristic in continual learning to mitigate catastrophic forgetting. As such, the method itself lacks significant innovation.
2. As the authors acknowledge, TALR offers negligible benefits over standard SFT when the recommended small learning rate is used. Its utility is confined to a niche scenario where one *must* use a large learning rate (to pursue peak in-domain performance) and needs to balance the resulting degradation in general capabilities.
3. The dynamic heuristic for setting the $\tau$ hyperparameter (using the batch median loss) is not sufficiently validated. The ablation study only compares it to a single fixed value ($\tau=1$) rather than a proper hyperparameter sweep or comparison with other dynamic strategies.
4. The experiment in Figure 3, which is used to motivate the "curriculum-like" behavior, does not match the actual TALR algorithm. The experiment uses gradient clipping (truncation to zero) for hard tokens, whereas TALR only uses down-weighting. Therefore, this analysis does not represent the true dynamics of TALR. Furthermore, the aggregate statistic in Figure 3c (the rising fraction of p>0.2 tokens) fails to prove that the hardest tokens (e.g., p=0.05) are eventually learned; it could simply reflect that medium-difficulty tokens have become easier.
5. All experiments are conducted on relatively small-scale models (8B or less). The paper lacks validation on whether its core conclusions generalize to current state-of-the-art large models (e.g., 70B+) or MoE architectures, whose fine-tuning dynamics may differ significantly.

**Questions:**

1. TALR balances performance by down-weighting hard tokens. As noted in the weaknesses, it is not well-argued whether these hard tokens are eventually learned. Could the authors provide an analysis that tracks the probability of a few specific, hard tokens (like the example in Fig 3a) throughout the training process to demonstrate their learning dynamics?
2. The paper lacks an analysis of the actual training dynamics of TALR, instead showing a related "extreme experiment" (Fig 3c). Could the authors provide an analysis (e.g., of token probabilities or loss weights) based on the true TALR algorithm as presented in Algorithm 1?
3. The paper's primary recommendation is to use a small learning rate, a setting where TALR offers minimal advantages. The utility of TALR hinges on a scenario where one must use a large LR to achieve peak domain performance. Could the authors provide a clearer experimental case where the domain performance of the small-LR SFT significantly lags behind that of the large-LR SFT, thereby strengthening the practical case for TALR?

---

> ### Author Response · Authors · 2025-11-21
>
> We sincerely thank the reviewer for the constructive feedback. Below we clarify the main contributions of the paper and respond to the specific concerns.
>
> **Main contribution of the paper**
>
> Our work has two tightly connected contributions.
> * (1) We revisit domain specific SFT for LLMs and show that SFT does not always hurt general capabilities. Using a smaller learning rate can substantially mitigate forgetting while achieving target domain performance that is comparable to that obtained with larger learning rates. This gives a better trade off between domain gains and general capabilities, which we support with both empirical and compression theoretic analysis. **[We will expand more about the discussion of novelty below.]**
> * (2) We further show that a small learning rate alone does not fully solve the problem, especially when stronger domain gains are desired and should using a larger learning rate. Motivated by our theoretical analysis, we derive TALR, a token adaptive reweighting scheme that directly reduces the impact of low probability tokens on forgetting.
>
> The paper is therefore not only an empirical and theoretical study of learning rate, and not only the introduction of TALR; it is a combined story in which the analysis of learning rate reveals a previously under appreciated regime for LLM SFT and then naturally leads to TALR as a practical mitigation method.
>
> ---
>
> **(1) Novelty of the small learning rate finding in LLMs’ domain-specific SFT**
>
> We would like to clarify that the central finding regarding small learning rates is not a re-statement of findings in continual learning on traditional machine/deep learing models but a phenomenon that is specific to LLMs under domain-specific SFT. Below are the details of our clarification:
>
> First, we agree that using a smaller learning rate to reduce catastrophic forgetting is expected, as we state in the introduction (lines 59–68). What is surprising in our setting is that, for LLMs under domain specific SFT, small learning rates can substantially mitigate forgetting while **still achieving comparable domain specific performance to much larger learning rates.**  Prior to the LLM era, practical experience in machine and deep learning suggested that larger learning rates are often essential for better downstream performance [see the citations in line 62-63 in our paper]. In contrast, we show that LLMs behave differently: comparable domain-specific performance can still be achieved under smaller learning rates. A key reason we hypothesize for the differences is that LLMs are high-capacity models, with large-scale parameters and trained on extremely vast amount of tokens, compared with traditional machine/deep learning models.
>
> To the best of our knowledge, this phenomenon has not been systematically studied or even clearly pointed out for LLMs. On the LLM side, **several recent papers including some published at top venues** like ICML 2025 and TMLR 2025 (for example those cited in lines 138–140) **report that domain specific SFT severely harms general capabilities** and largely treat this as an unavoidable consequence. These works typically adopt relatively large learning rates and **do not investigate the small learning rate regime** or the possibility of achieving comparable domain performance while mitigating forgetting. We would also greatly appreciate it if the reviewer could point us to prior papers on LLMs that already establishes the same phenomenon.
>
> Our contribution is to demonstrate that LLMs behave differently from conventional models in this respect and to provide both empirical evidence and theoretical justification for it. Given the growing practice of fine tuning LLMs on domain data in university labs and in industry settings with limited resources, we believe that **making this regime explicit and providing clear guidelines is practically important for the community.**
>
> **(2) On the practical regime where TALR is useful**
>
> We appreciate the reviewer’s request for a clearer experimental case where small learning rate SFT underperforms large learning rate SFT on the target domain. We make this distinction more explicit by presenting the Figure 2 results in table form. As shown in the tables below, when moving from a learning rate of 1e-6 to 5e-6, the average domain performance improves notably across all three models, confirming precisely the situations where the domain performance of the small-LR SFT significantly lags behind that of the large-LR SFT.

---

> ### Author Response · Authors · 2025-11-21
>
> **Performance Summary Table (LR=1e-6)**
>
> **Table 1: Results at Learning Rate 1e-6. Highlighting the Best General Performance Among Methods Matching Standard SFT in Domain Accuracy**
> |Method|Qwen2.5-3BDomain|Qwen2.5-3BGeneral|Qwen3-4BDomain|Qwen3-4BGeneral|Gemma3-4BDomain|Gemma3-4BGeneral|AvgDomain|AvgGeneral|
> |------|----------------|------------------|--------------|----------------|----------------|------------------|----------|-----------|
> |Standard|0.4947|0.6202|0.5484|0.7837|0.5587|0.6734|0.5339|0.6924|
> |L2-Reg|0.4904|0.6205|0.4692|0.7964|0.5595|0.6750|0.5064|0.6973|
> |LoRA|0.1261|0.5831|0.1945|0.7640|0.2233|0.1241|0.1813|0.4904|
> |Wise-FT|0.1948|0.6285|0.1428|0.7884|0.2573|0.7635|0.1983|0.7268|
> |FLOW|0.3641|0.5974|0.4768|0.7870|0.5673|0.6914|0.4694|0.6920|
> |TALR|0.4806|0.6478|0.4889|0.7880|0.5338|0.7150|0.5011|**0.7169**|
>
> **Performance Summary Table (LR=5e-6)**
> | Method | Qwen2.5-3B Domain | Qwen2.5-3B General | Qwen3-4B Domain | Qwen3-4B General | Gemma3-4B Domain | Gemma3-4B General | Avg Domain | Avg General |
> |--------|-------------------|--------------------|-----------------|--------------------|------------------|-------------------|------------|-------------|
> | Standard | 0.5459 | 0.3337 | 0.5782 | 0.5425 | 0.5507 | 0.2655 | 0.5583 | 0.3805 |
> | L2-Reg | 0.5406 | 0.3470 | 0.5782 | 0.5591 | 0.5471 | 0.2796 | 0.5553 | 0.3952 |
> | LoRA | 0.1734 | 0.5670 | 0.2367 | 0.7571 | 0.3864 | 0.1241 | 0.2655 | 0.4827 |
> | Wise-FT | 0.3584 | 0.5869 | 0.3815 | 0.7531 | 0.4638 | 0.5929 | 0.4012 | 0.6443 |
> | FLOW | 0.5266 | 0.4419 | 0.5819 | 0.5599 | 0.5500 | 0.3476 | 0.5528 | 0.4498 |
> | TALR | 0.5066 | 0.5490 | 0.5834 | 0.6138 | 0.5351 | 0.3427 | 0.5417 | **0.5018** |
>
> From the 5e-6 table, one can see that TALR attains domain performance that is comparable to standard SFT in terms of average domain score, while at the same time **substantially improving average general performance by 6–12 points over the baselines.** This demonstrates that TALR is a useful tool in the realistic setting where practitioners seek the stronger domain gains enabled by a larger learning rate yet still wish to preserve general capabilities as much as possible. We will emphasize this scenario more clearly in the main text.
>
> **(3) About whether hard tokens are learned or not**
>
> Thank you for raising this concern. To directly address whether hard tokens are eventually learned, we conducted an additional analysis on Qwen2.5-3B-Instruct trained on MedCalc. We track, throughout training, the fraction of tokens whose predicted probability satisfies p<0.05, which we use as an operational definition of “hard tokens.” The results are shown in Figure 8 in the appendix C.7.
>
> We observe that during the first epoch, the proportion of such low-probability tokens decreases rapidly, and by the second epoch it approaches zero and stays near zero. This behavior demonstrates that the hard tokens are gradually learned by the model during training. These learning dynamics confirm that the model does learn the hard tokens, and TALR’s down-weighting does not prevent them from being learned.
>
> **(4) Learning Dynamics of TALR**
>
> Thank you for the helpful suggestion. To directly analyze the learning dynamics under the true TALR algorithm, we track three curves during training on Qwen2.5-3B-Instruct with MedCalc:
> the token loss before reweighting, the training loss after reweighting (the actual training loss), and the dynamic hyperparameter τ used by TALR. The results are shown in Figure 9.
>
> We observe that both token losses and the final training loss decrease rapidly during the first epoch and continue to stabilize during the second epoch. Moreover, τ also decreases steadily throughout training. Combined with all these learning curves, this reflects that most of tokens initially considered hard become easier over time. These curves confirm that TALR allows the model to learn normally.

---

> ### Author Response · Authors · 2025-11-21
>
> **(5) More ablation study of hyperparameter** $\tau$
>
> We thank the reviewer for raising this point. We conducted additional ablation studies to more thoroughly evaluate the choice of τ, including both fixed values and alternative dynamic strategies. The expanded results are shown below.
>
> Our findings can be summarized as follows.
> * First, the fixed-τ baselines perform noticeably worse than the dynamic τ used in our main experiments. This gap becomes especially pronounced under the larger learning rate (5e-6). This confirms that a single manually chosen τ is not robust across settings.
> * Second, we evaluated an alternative dynamic strategy, which we refer to as “median max” to compare with our paper’s dynamic way “median avg”, i.e. the median of average sequence losses in each batch. However, as shown in the table, median max consistently underperforms the “median avg”, both in domain and general performance. Among these ablation studies, the median avg variant yields the most stable behavior across learning rates and achieves the strongest overall trade-off.
>
> | τ   | Domain (1e-6) | General (1e-6) | Avg (1e-6) | Domain (5e-6) | General (5e-6) | Avg (5e-6) |
> |-----------|----------------|-----------------|------------|----------------|-----------------|------------|
> | 0.01      | 0.1977         | 0.6060          | 0.4019     | 0.1193         | 0.5735          | 0.3464     |
> | 0.5       | 0.2330         | 0.5287          | 0.3809     | 0.1241         | 0.6019          | 0.3630     |
> | 2         | 0.3381         | 0.5593          | 0.4487     | 0.5147         | 0.4743          | 0.4945     |
> | 5         | 0.5721         | 0.6631          | 0.6176     | 0.5978         | 0.2736          | 0.4357     |
> | Median max| 0.4202         | 0.6686          | 0.5444     | 0.6103         | 0.4822          | 0.5462     |
> | Median avg (in our original submission)| 0.5033         | 0.7833          | **0.6433**     | 0.5988         | 0.5847          | **0.5918**  |
>
> **(6) On model scale**
>
> We acknowledge the reviewer’s concern regarding model scale, which is also mentioned in our paper’s Limitations section. Our experiments are conducted on models of size up to 8B parameters due to compute resource constraints. In practice, many university labs and a substantial fraction of industry deployments with constrained compute budgets fine tune models in the 1B–8B range, especially for domain specific applications such as medical, e-commerce or industry internal data. Our findings therefore target a regime that is highly relevant for these users and provide concrete, actionable guidance for their SFT workflows. At the same time, we agree that understanding whether the same phenomena extend to much larger dense models or MoE architectures is an important open question, leaving as future work.
>
> ---
>
> We thank the reviewer again for your valuable comments. We hope these clarifications address your concerns and highlight the scientific value of our work.

---

> > ### Author Response · Authors · 2025-11-29
> >
> > Inspired by Reviewer mWRP’s insightful question, we further investigated the practical value of TALR, especially under larger learning rate settings. Specifically, we surveyed the supervised fine-tuning (SFT) learning rates used in representative open-source LLMs released between 2023 and 2025. The results are summarized below:
> >
> > **Table: SFT Learning Rates in Representative LLMs**
> > | LLM | SFT learning rate | Year |
> > |------|----------------------|------|
> > | LLaMA-2 | 2e-5 | 2023 |
> > | Mistral | Not disclosed | 2023 |
> > | LLaMA-3 | 1e-5 | 2024 |
> > | DeepSeek-V3 | 5e-6 | 2024 |
> > | Qwen-2.5 | 7e-6 | 2024 |
> > | Qwen-3 | Not disclosed | 2025 |
> > | Olmo-3 | 5e-5 | 2025 |
> >
> > Two conclusions follow:
> >
> > (1) The majority of widely used LLMs adopt relatively large SFT learning rates. Since TALR performs more prominently under high learning rate settings, TALR is practically important: it improves performance exactly in the LR regimes that modern LLMs actually use.
> >
> > (2) Our finding that small learning rates offer a better trade-off is our contribution. Since many representative LLMs use larger LRs, our paper shows that:
> > * small LR yields a better domain–general trade-off than these widely used defaults
> > * and this phenomenon had not been previously identified or systematically studied for LLMs
> >
> > ---
> >
> > Together, these findings strengthen our contributions and the practical significance of TALR :
> > TALR not only offers improvements where current LLMs most commonly use larger learning rates, but our analysis also reveals that the community may benefit from reconsidering the default LR choices altogether.

---

### Official Review · Reviewer_mWRP · 2025-10-28

**Soundness:** 3
**Presentation:** 3
**Contribution:** 2
**Rating:** 4
**Confidence:** 4

**Summary:**

Based on experiments showing that a smaller learning rate can partially mitigate catastrophic forgetting, this paper validates the conclusion through theoretical analysis. Furthermore, building on both theoretical and experimental findings, the authors propose the TALR method. By limiting the magnitude of updates for low-probability tokens, the approach alleviates catastrophic forgetting to some extent.

**Strengths:**

1. This paper is well-written and really easy to read and follow.

2. The method proposed in this paper is derived from an experimental finding and theoretical analysis, with a well-justified motivation.

**Weaknesses:**

1. I think the paper's core motivation, using a smaller learning rate to mitigate catastrophic forgetting, is clear. However, a significant issue is that this finding is not novel, as it is already a widely recognized common-sense approach. The paper excessively emphasizes this observation, dedicating substantial space to elaborating the advantages of a low learning rate, while its actual contribution, the proposed TALR method, does not receive sufficient attention or elaboration. This may lead readers to perceive the paper's main contribution as the experimental and theoretical analysis of the low learning rate's advantages, thereby overlooking the more important innovation.

2. Building on the previous point, if the main contribution of this paper is merely a more thorough experimental and theoretical analysis of the low learning rate, I do not believe the contribution is sufficient, as the experimental results do not present any particularly striking or novel conclusions.

3. Regarding Figure 2, could you provide the results in a table format? From the figure, it appears that TALR does not show a clear advantage under many settings, which suggests that the TALR method may not have a significant effect.

4. Regarding the extensive discussion on low learning rates in the paper, I believe the experimental analysis is not sufficiently comprehensive. In both the preliminary analysis and the main experiments, the authors only used three different learning rate settings (with even just two shown in Figure 2). Since TALR is derived from this experimental finding, I think the experiments should be more thorough and robust—relying on only three learning rate points is inadequate.

**Questions:**

What is the main contribution of this paper? Is it the experimental and theoretical analysis of the low learning rate, or is it the TALR method?

---

> ### Author Response · Authors · 2025-11-21
>
> We sincerely thank the reviewer for the constructive feedback. Below we first clarify the main contributions of the paper and respond to the specific concerns.
>
> **Main contribution of the paper**
>
> Our work has two tightly connected contributions.
> * (1) We revisit domain specific SFT for LLMs and show that SFT does not always hurt general capabilities. Using a smaller learning rate can substantially mitigate forgetting while achieving target domain performance that is comparable to that obtained with larger learning rates. This gives a better trade off between domain gains and general capabilities, which we support with both empirical and compression theoretic analysis. **[We will expand more about the discussion of novelty below.]**
> * (2) We further show that a small learning rate alone does not fully solve the problem, especially when stronger domain gains are desired and should using a larger learning rate. Motivated by our theoretical analysis, we derive TALR, a token adaptive reweighting scheme that directly reduces the impact of low probability tokens on forgetting.
>
> The paper is therefore not only an empirical and theoretical study of learning rate, and not only the introduction of TALR; it is a combined story in which the analysis of learning rate reveals a previously under appreciated regime for LLM SFT and then naturally leads to TALR as a practical mitigation method.
>
> ---
>
> **(1) Novelty of the small learning rate finding in LLMs’ domain-specific SFT**
>
> We would like to clarify that the central finding regarding small learning rates is not a re-statement of findings in continual learning on traditional machine/deep learning models but a phenomenon that is specific to LLMs under domain-specific SFT. Below are the details of our clarification:
>
> First, we agree that using a smaller learning rate to reduce catastrophic forgetting is expected, as we state in the introduction (lines 59–68). What is surprising in our setting is that, for LLMs under domain specific SFT, small learning rates can substantially mitigate forgetting **while still achieving comparable domain specific performance to much larger learning rates**.  Prior to the LLM era, practical experience in machine and deep learning suggested that larger learning rates are often essential for better downstream performance [see the citations in line 62-63 in our paper]. In contrast, we show that LLMs behave differently: comparable domain-specific performance can still be achieved under smaller learning rates. A key reason we hypothesize for the differences is that LLMs are high-capacity models, with large-scale parameters and trained on extremely vast amount of tokens, compared with traditional machine/deep learning models.
>
> To the best of our knowledge, this phenomenon has not been systematically studied or even clearly pointed out for LLMs. On the LLM side, **several recent papers including some published at top venues** like ICML 2025 and TMLR 2025 (for example those cited in lines 138–140) **report that domain specific SFT severely harms general capabilities** and largely treat this as an unavoidable consequence. These works typically adopt relatively large learning rates and do not investigate the small learning rate regime or the possibility of achieving comparable domain performance while mitigating forgetting. We would also greatly appreciate it if the reviewer could point us to prior papers on LLMs that already establish the same phenomenon as our paper.
>
> Our contribution is to demonstrate that LLMs behave differently from conventional models in this respect and to provide both empirical evidence and theoretical justification for it. Given the growing practice of fine tuning LLMs on domain data in university labs and in industry settings with limited resources, we believe that **making this regime explicit and providing clear guidelines is practically important for the community.**
>
>
> **(2) Figure 2 and table format**
>
> We thank the reviewer for the helpful suggestion. Following your request, we convert Figure 2 into table form below. This presentation makes it easier to inspect the exact numerical differences across methods. From these tables, we draw two observations.
>
> First, for the smaller learning rate (1e-6), standard SFT already mitigates general-performance degradation while maintaining strong domain accuracy, consistent with our earlier analysis.
>
> Second, and more importantly, for the larger learning rate (5e-6), TALR achieves the strongest trade-off among all methods. While maintaining domain performance comparable to standard SFT, TALR substantially reduces forgetting, **improving average general performance by 6–12 points over other baselines**. This confirms the effectiveness of our proposed TALR, offering an even better balance between domain gains and general capability retention for larger learning rates.
>
> Below are detailed analysis.

---

> > ### Author Response · Authors · 2025-11-21
> >
> > **Performance Summary Table (LR=1e-6)**
> >
> > For the 1e-6 learning rate, the tables confirm our main claim stated in the paper: simply using a small learning rate already mitigates general-performance degradation while preserving strong domain performance. However, 1e-6 also limits domain performance, which motivates the use of a slightly larger learning rate such as 5e-6 to further improve domain accuracy.
> >
> > **Table 1: Results at Learning Rate 1e-6. Highlighting the Best General Performance Among Methods Matching Standard SFT in Domain Accuracy**
> > | Method   | Qwen2.5-3B Domain | Qwen2.5-3B General | Qwen3-4B Domain | Qwen3-4B General | Gemma3-4B Domain | Gemma3-4B General | Avg Domain | Avg General |
> > |----------|--------------------|---------------------|------------------|-------------------|-------------------|--------------------|------------|-------------|
> > | Standard | 0.4947             | 0.6202              | 0.5484           | 0.7837            | 0.5587            | 0.6734             | 0.5339     | 0.6924      |
> > | L2-Reg   | 0.4904             | 0.6205              | 0.4692           | 0.7964            | 0.5595            | 0.6750             | 0.5064     | 0.6973      |
> > | LoRA     | 0.1261     | 0.5831              | 0.1945           | 0.7640            | 0.2233            | 0.1241             | 0.1813     | 0.4904      |
> > | Wise-FT  | 0.1948        | 0.6285              | 0.1428           | 0.7884            | 0.2573            | 0.7635             | 0.1983     | 0.7268      |
> > | FLOW     | 0.3641    | 0.5974              | 0.4768           | 0.7870            | 0.5673            | 0.6914             | 0.4694     | 0.6920      |
> > | TALR     | 0.4806     | 0.6478              | 0.4889           | 0.7880            | 0.5338            | 0.7150             | 0.5011     | **0.7169**      |
> >
> >
> >
> >
> > **Performance Summary Table (LR=5e-6)**
> >
> > When comparing 5e-6 results across methods, two patterns become clear.
> > * First, as expected, moving from 1e-6 to 5e-6 consistently improves domain performance across all models, validating our discussion of scenarios where a larger learning rate is needed for stronger target-task performance.
> > * Second, TALR achieves the best overall trade-off, achieving comparable domain performance and significantly mitigate forgetting.
> >
> > At both learning rates, TALR maintains comparable domain performance to the best baselines (see “Avg Domain”), and its Avg General performance is the highest among the baselines that achieve similar domain performance, outperforming other baselines by 6–12 points. This verifies the effectiveness of our proposed TALR method.
> >
> > **Table 2: Results at Learning Rate 5e-6. Highlighting the Best General Performance Among Methods Matching Standard SFT in Domain Accuracy**
> > | Method | Qwen2.5-3B Domain | Qwen2.5-3B General | Qwen3-4B Domain | Qwen3-4B General | Gemma3-4B Domain | Gemma3-4B General | Avg Domain | Avg General |
> > |--------|-------------------|--------------------|-----------------|--------------------|------------------|-------------------|------------|-------------|
> > | Standard | 0.5459 | 0.3337 | 0.5782 | 0.5425 | 0.5507 | 0.2655 | 0.5583 | 0.3805 |
> > | L2-Reg | 0.5406 | 0.3470 | 0.5782 | 0.5591 | 0.5471 | 0.2796 | 0.5553 | 0.3952 |
> > | LoRA | 0.1734 | 0.5670 | 0.2367 | 0.7571 | 0.3864 | 0.1241 | 0.2655 | 0.4827 |
> > | Wise-FT | 0.3584 | 0.5869 | 0.3815 | 0.7531 | 0.4638 | 0.5929 | 0.4012 | 0.6443 |
> > | FLOW | 0.5266 | 0.4419 | 0.5819 | 0.5599 | 0.5500 | 0.3476 | 0.5528 | 0.4498 |
> > | TALR | 0.5066 | 0.5490 | 0.5834 | 0.6138 | 0.5351 | 0.3427 | 0.5417 | **0.5018** |
> >
> > **(3) Number of learning rate settings**
> >
> > We thank the reviewer for raising this point. In response, we have expanded our experiments to include a more fine-grained sweep of learning rate values. Specifically, we conduct an additional study on Qwen3-8B with the MedCalc dataset, evaluating six different learning rates.
> >
> > The expanded sweep confirms the same empirical pattern reported in the main paper. In particular, a smaller learning rate (1e-6) achieves domain performance comparable to larger learning rates while substantially mitigating forgetting on general-purpose benchmarks. This reinforces our main finding that small learning rates yield a more favorable trade-off for domain-specific SFT. These additional results strengthen the robustness of our conclusions.
> >
> > |LearningRate|DomainPerformance|GeneralPerformance|
> > |------------|-----------------|------------------|
> > |1e-6|0.6122|0.7734|
> > |2e-6|0.6227|0.6290|
> > |5e-6|0.6074|0.4788|
> > |8e-6|0.5969|0.3226|
> > |1e-5|0.5940|0.3174|
> > |2e-5|0.5434|0.0945|
> >
> > ---
> >
> > Once again, we thank the reviewer for the thoughtful comments. We will incorporate the above clarifications, the new learning rate sweep, and the tabular presentation of Figure 2 into the revised version to better highlight both the learning rate analysis and the TALR method.

---

> > > ### Comment · Reviewer_mWRP · 2025-11-28
> > >
> > > Thank you for the authors' response. I have one final question that I hope to discuss with the authors, and I will raise my score if I believe this last question is addressed.
> > >
> > > Regarding the results in Table 1: In your response, you emphasized that a low learning rate can mitigate catastrophic forgetting while maintaining a high level of downstream performance. In Table 1, the "Standard" method achieves an Avg Domain score of 53.39 and an Avg General score of 69.24, whereas TALR yields an Avg Domain score of 50.11 and an Avg General score of 71.69. My question is: Under the low learning rate setting, although TALR demonstrates better forgetting mitigation, is it necessary to sacrifice 3 percentage points in downstream task performance to gain only 2 percentage points in generalizability?
> > >
> > > While the results in Table 2 show that TALR performs more prominently under high learning rate settings, to my knowledge, current post-training of large language models generally adopts low learning rates. For example, the Qwen 2.5 technical report uses an initial learning rate of 1e-6. Therefore, under the low learning rate setting in Table 1, is it justifiable to sacrifice more downstream task performance for improved generalizability?

---

> > > > ### Author Response · Authors · 2025-11-29
> > > >
> > > > We sincerely thank the reviewer for the follow-up question. The discussion you raise is valuable. Below we clarify two key points with additional evidence.
> > > >
> > > > <1> Empirical SFT learning rates in modern LLM post-training are **not** dominated by small LR such as 1e-6
> > > >
> > > > We appreciate the reviewer’s comment regarding current SFT post-training recipes and therefore conducted a survey of representative open-source LLMs (2023–2025).
> > > >
> > > > The SFT learning rates used in practice are shown below:
> > > >
> > > > **Table: SFT Learning Rates in Representative LLMs**
> > > > | LLM | SFT learning rate | Year |
> > > > |------|----------------------|------|
> > > > | LLaMA-2 | 2e-5 | 2023 |
> > > > | Mistral | Not disclosed | 2023 |
> > > > | LLaMA-3 | 1e-5 | 2024 |
> > > > | DeepSeek-V3 | 5e-6 | 2024 |
> > > > | Qwen-2.5 | 7e-6 | 2024 |
> > > > | Qwen-3 | Not disclosed | 2025 |
> > > > | Olmo-3 | 5e-5 | 2025 |
> > > >
> > > > Two conclusions follow:
> > > >
> > > > (1) **The majority of widely used LLMs adopt relatively large SFT learning rates**. Moreover, Qwen-2.5 does **not** use 1e-6 **but 7e-6**. Given this, our and the reviewer’s own observation “TALR performs more prominently under high learning rate settings” is precisely explaining why TALR is practically important: it improves performance exactly in the LR regimes that modern LLMs actually use.
> > > >
> > > > (2) Our finding that **small learning rates offer a better trade-off is our contribution**. Since many representative LLMs use larger LRs, our paper shows that:
> > > > * smaller LR yields a better domain–general trade-off than these widely used defaults
> > > > * and this phenomenon had not been previously identified or systematically studied for LLMs
> > > >
> > > > <2>
> > > >
> > > > With the above clarifications in place, we now revisit Table 1 to directly respond to the reviewer’s question.
> > > >
> > > > (a) Both “Standard SFT (small LR)” and “TALR” are our contributions and lie on the Pareto frontier among all baselines. Practitioners may choose either depending on whether they prioritize domain accuracy or general capabilities.
> > > >
> > > > (b) These trade-offs depend on models and tasks. In our results, small-LR SFT and TALR consistently emerge as the two best-performing trade-off options, and **practitioners can select either depending on their goals**. In most cases, TALR offers the more favorable balance.
> > > >
> > > > (c) We observed that Qwen3-4B (https://huggingface.co/Qwen/Qwen3-4B) is an outlier in table 1 (1e-6 results). Specifically, Qwen3-4B showed unusually high standard-SFT domain performance vs L2-reg and unstable learning curves (sudden rise/drop). This anomaly does not appear in any of the other 5 settings and models in Table 1 & 2 in our previous response. We hypothesized that the “Qwen/Qwen3-4B” might have issues.
> > > >
> > > > Indeed, in the latter half of this year, the Qwen team released Qwen3-4B-Instruct-2507 (https://huggingface.co/Qwen/Qwen3-4B-Instruct-2507), an updated version.
> > > > We reran the same experiment (1e-6 LR + MedCalc):
> > > >
> > > > | Method | Domain Performance | General Performance |
> > > > |--------|---------------------|----------------------|
> > > > | Standard SFT | 0.5988 | 0.8314 |
> > > > | TALR | 0.5864 | 0.8531 |
> > > >
> > > > On the updated model, Standard SFT and TALR again both lie on the Pareto frontier, and TALR maintains general capability and archives comparable domain performance, addressing the reviewer’s concern.
> > > >
> > > > ---
> > > >
> > > > To summarize:
> > > > * By checking representative LLM post-training recipes, we found that most modern LLMs use larger SFT learning rates → TALR is highly relevant in the LR regimes that modern LLMs actually use.
> > > > * Small-LR SFT and TALR are both our contributions and  strong Pareto-optimal trade-off choices.
> > > > * TALR provides a clean improvement in general capability with comparable domain performance.
> > > >
> > > > We hope this clarifies the reviewer’s concern, and we sincerely appreciate the thoughtful follow-up question.

---

### Official Review · Reviewer_9brq · 2025-10-30

**Soundness:** 3
**Presentation:** 3
**Contribution:** 3
**Rating:** 6
**Confidence:** 3

**Summary:**

This paper challenges the common belief that domain-specific supervised fine-tuning (SFT) significantly harms the general capabilities of large language models (LLMs). Through empirical and theoretical analysis, the authors demonstrate that using a smaller learning rate can mitigate this performance degradation while maintaining strong domain-specific performance. They further propose a novel method, Token-Adaptive Loss Reweighting (TALR), which dynamically down-weights the loss of "hard" tokens during training to better preserve general capabilities.

**Strengths:**

1. The writing is experienced and clear, effectively motivating the problem and delivering its overall message.

2. The work revisits a critical topic—the trade-off between in-distribution and out-of-distribution performance during SFT. Its core finding, that smaller learning rates are a simple yet effective strategy, is a valuable and practical insight for practitioners.

3. The token-level analysis is a particular strength, yielding interesting findings. For instance, it shows that most tokens in SFT data are easy for the model, and performance bottlenecks are likely due to a sparse set of hard tokens.

4. The theoretical analysis, framed through an information-theoretic and compression-based lens, is interesting and provides a valid foundation to explain the empirical phenomena observed, such as why smaller learning rates help.

**Weaknesses:**

1. Figure 2 can be difficult to interpret in its current form. Splitting it into more straightforward visualizations, such as bar charts, could make the conclusions more visually clear and accessible.

2. The advantage of the proposed TALR method over baselines is not entirely consistent. At a learning rate of 1e-6, methods like L2 regularization perform on par with TALR. TALR shows a clearer advantage at 5e-6 on Qwen models, but this benefit does not consistently extend to the Gemma models. This suggests that the effectiveness of mitigation strategies may be dependent on the nature or distribution of the base model. Authors could provide more explanations/explorations on this angle

**Questions:**

1. In Figure 1, for some models (e.g., Qwen2.5-7B on ESCI w/ CoT), domain-specific SFT appears to slightly improve general performance compared to the initialization. Could the authors provide hypotheses or explanations for this unexpected improvement?

2. The appendix states that 20 training epochs were set for MedCalc and ESCI, which seems high for domain-specific SFT. What was the rationale for this choice? Providing a plot of performance versus epochs for key models and learning rates would be helpful. This would clarify whether models fine-tuned with large learning rates might have been overtrained, potentially confounding the results.

---

> ### Author Response · Authors · 2025-11-21
>
> We sincerely thank the reviewer for the constructive feedback, the positive assessment of our work, and the thoughtful suggestions. We appreciate your recognition of the clarity of the writing, the significance of the token-level analysis, and the value of our theoretical and empirical findings. Below we respond to your specific questions and concerns.
>
> 1. Thank you for pointing this out. We provide several hypotheses that may help explain the slight improvement in general performance for some settings. The domain data used in these cases adopts a chain-of-thought format that contains reasoning-oriented and structured explanations. These CoT-style rationales share characteristics with general evaluation datasets that involve step-by-step reasoning, lightweight logical inference, or structured explanatory patterns. As a result, supervised fine-tuning with a small learning rate, or a small update step, on such data can inadvertently act as a mild enhancement of instruction-following and reasoning ability. This can lead to small but observable improvements on general benchmarks.
>
> 2. The choice of twenty epochs for ESCI and MedCalc was primarily intended to obtain a more complete and reliable view of how different learning rates influence both domain performance and general performance. This design choice was not meant to imply that practical SFT requires training for twenty epochs, **but rather to ensure that our analysis covers sufficiently many optimization steps for meaningful comparisons across conditions and situations**. We have added performance versus epoch curves to Appendix C.6. These plots reveal several important patterns. For example, for Gemma3-4B at a learning rate of 1e-6, the domain performance reaches its peak only around epoch 12, indicating that extended training can meaningfully benefit domain-specific performance when the learning rate is small. For larger learning rates, the performance peak also appears in later epochs, suggesting that the degradation in general capabilities is primarily determined by the learning rate magnitude itself rather than by over-training.
>
> Moreover, although the training was configured to run for up to twenty epochs, our evaluation protocol prevents over-training from affecting the reported results. As described in Section 3.1.2, we follow prior work in selecting checkpoints based on their best performance on the target domain task, after which the corresponding checkpoints are evaluated on general-purpose benchmarks. This setup reflects practical usage scenarios where downstream task performance is prioritized and ensures that our comparisons are made at matched points of maximal domain performance rather than at arbitrary late-stage checkpoints.
>
> ---
>
> We thank the reviewer again for your valuable comments. We hope these clarifications address your concerns and highlight the scientific value of our work. We will keep updating our manuscript and will revise it based on your comments.

---

### Official Review · Reviewer_XUav · 2025-10-31

**Soundness:** 3
**Presentation:** 3
**Contribution:** 3
**Rating:** 6
**Confidence:** 3

**Summary:**

The paper shows that domain-specific SFT does not necessarily hurt LLM general abilities: small learning rates suffice to retain them while keeping domain gains. It gives a compression-theoretic explanation and introduces TALR, an adaptive token re-weighting method that further curbs forgetting.

**Strengths:**

- Revisits and refutes a prevailing assumption—namely, that domain-specific SFT necessarily degrades general capabilities—showing instead that optimization choices, particularly learning-rate magnitude, are the primary determinant of forgetting.

- Employ learning rate as a single, easily tunable knob that largely eliminates catastrophic forgetting without extra parameters or data.

- TALR is lightweight (closed-form, no extra forward passes), yet consistently outperforms recent regularizers and MoE-style baselines.

**Weaknesses:**

- Scope limited to decoder-only LLMs ≤8 B; no evidence that conclusions hold for larger or encoder-decoder models.

- No probing of emergent skills (e.g., long-context) or fairness/toxicity side-effects after SFT.

**Questions:**

- Did you experiment with top-k or sparse routing variants of TALR, and how do they affect convergence speed and final trade-offs?

- How does TALR behave when the domain corpus is large-scale (>>1 M examples) or when continual domain increments arrive sequentially—does τ need scheduling?

---

> ### Author Response · Authors · 2025-11-21
>
> We sincerely thank the reviewer for the constructive feedback and thoughtful suggestions. We are particularly encouraged that reviewers appreciated our paper. Below we respond to your specific questions.
>
> 1. Thank you for this insightful suggestion. Motivated by your question, we conducted an additional ablation study to evaluate sparse variants of TALR.
>
> We experimented with Qwen2.5-3B-Instruct on the MedCalc dataset using two variants:
> - (1) Hard top-p style filtering: At each optimization step, we only update tokens whose model probability satisfies p > 0.01. Result on MedCalc: 21.78.
> - (2) Sparse reweighting: For tokens with p < 0.01, instead of zeroing them out, we multiply their learning-rate weight by a fixed constant 0.01. Result on MedCalc: 31.52.
>
> In comparison, TALR achieves 50.14 under the same setup. These results show that aggressively sparsifying the update,, substantially reduces downstream domain performance. This observation aligns with our theoretical intuition that a soft and continuous reweighting over all tokens allows TALR to retain useful but low-probability signals while still mitigating forgetting. Overall, the current TALR formulation remains the most effective among the variants we tested.
>
> 2. We appreciate the reviewer’s question regarding data scalability. We partially address this in Appendix C.3, where we evaluate TALR on MetaMathQA, a corpus of roughly 0.4 million examples, which is substantially larger than MedCalc and ESCI. We find that our main findings continue to hold.
>
> 3. On sequential domain-specific SFT with new domain increments
>
> We thank the reviewer for raising this important question about the behavior of our method when domain data arrive sequentially. To examine this setting, we conducted an additional experiment using Qwen3-8B, where the model is first fine-tuned on MedCalc and then further fine-tuned on ESCI. The results are below.
>
> After training on MedCalc alone, the model achieves a MedCalc score of 0.5616 and a general benchmark score of 0.7734. We then continue training on ESCI. After this second stage, the MedCalc performance remains essentially unchanged at 0.5597, and the general benchmark score stays almost identical at 0.7673. Meanwhile, the model reaches a score of 0.5827 on ESCI.
>
> These results indicate that performance on the first domain is largely preserved, even after additional fine-tuning on a new domain, and that general capabilities do not suffer noticeable degradation. This sequential setting therefore further supports the effectiveness and robustness of our approach under continual domain expansion.
>
> 4. On model scale
>
> We acknowledge the reviewer’s concern regarding model scale, which is also mentioned in our paper’s Limitations section. Our experiments are conducted on models of size up to 8B parameters due to compute resource constraints. In practice, many university labs and a substantial fraction of industry deployments with constrained compute budgets finetune decoder-only models in the 1B–8B range, especially for domain specific applications such as medical or enterprise internal data. Our findings therefore target a regime that is highly relevant for these users and provide concrete, actionable guidance for their SFT workflows. At the same time, we agree that understanding whether the same phenomena extend to much larger models or encoder-decoder models is an important open question, leaving it as future work.
>
> ---
>
> We thank the reviewer again for your valuable comments. We hope these clarifications address your concerns and highlight the scientific value of our work.

---

> > ### Comment · Reviewer_XUav · 2025-11-27
> >
> > Thanks for the response. I do hope authors can take weaknesses into the revision carefully. I will adjust my rating accordingly.

---

> > > ### Author Response · Authors · 2025-11-29
> > >
> > > Thank you so much for raising the score and for your constructive feedback. We appreciate your support and will update our paper accordingly in the final manuscript.

---

### Author Response · Authors · 2025-11-30
**Author Final Remarks by Authors**

We sincerely thank the reviewers and ACs for their time, constructive feedback, and thoughtful engagement. We are encouraged by the recognition of our practical insights on smaller learning rates for achieving better SFT trade-off (Reviewer XUav, 9brq, T4Nt), strength of our empirical and theoretical analysis (mWRP, T4Nt), the novelty of our token-level analysis perspective (9brq), and the well-motivated design of TALR (XUav, 9brq, T4Nt).

_**Key Contributions**_

**1. Revisiting a widely held assumption in recent LLM literature.**
Several influential works published at top venues (e.g., ICML, TMLR) have reported that supervised fine-tuning (SFT) causes substantial catastrophic forgetting in LLMs.
Our study systematically re-examines this claim and reveals a new phenomenon specific to LLMs: small learning rates can substantially mitigate forgetting while still achieving domain-specific performance comparable to that of much larger learning rates.
This identifies a previously unrecognized better trade-off regime and provides actionable guidance for practitioners.

**2. A theoretically grounded and empirically effective method: TALR.** Building upon our theoretical analysis, we introduce Token-Adaptive Loss Reweighting (TALR), which adaptively down-weights the updates on hard/low-probability tokens to preserve general capabilities.
TALR is simple, lightweight, closed-form, and consistently outperforms prior baselines, especially in the large-LR regime widely used in modern post-training pipelines.

---

_**Rebuttal Updates Summary**_

**1. Novelty of the small-LR phenomenon (Reviewers mWRP, T4Nt)**

In the discussion phase, we clarified that our central finding is not a re-statement of conventional continual-learning findings from traditional machine/deep learning.
Instead, it is a LLM-specific phenomenon that emerges under domain-specific SFT.
Past LLM studies did not examine the small-LR regime, and in fact reported strong degradation while using large LRs.
Our empirical and theoretical analysis fills this gap and establishes a new understanding of SFT dynamics for LLMs.

**2. Figure 2 presentation improvements (Reviewers 9brq, mWRP)**

Following the reviewers’ suggestion, we redesigned Figure 2 using tabular presentation (Table 1 and Table 2 in revised manuscript) for clarity.
The tables clearly show that:
* At 1e-6, small-LR SFT and TALR (both our contributions) achieve the best trade-offs among all baselines.
* At 5e-6, TALR shows clear and significant superiority, achieving comparable domain performance while mitigating forgetting by 6–12 points.

**3. Current post-training recipes in modern LLMs (Reviewer mWRP)**

Motivated by the reviewer mWRP’s insightful question, we conducted a survey of SFT learning rates used in representative LLMs (e.g., LLaMA, Qwen, DeepSeek, Olmo).
We found that modern post-training pipelines use relatively large learning rates. This leads to two implications:
* Our finding that small LRs yield a better trade-off is an important and previously overlooked contribution, since these representative LLM SFT recipes typically operate in the large-LR regime.
* TALR’s effectiveness in large-LR settings becomes practically important, as it improves general capability exactly in the LR range that modern LLMs actually use.

---

_**Overall Impact**_

In summary, our work (1) revises the community’s understanding of SFT-induced forgetting in LLMs, (2) introduces a simple and theoretically grounded method TALR for mitigating degradation, and (3) provides practical guidance directly applicable to current LLM post-training practices. We believe these contributions meaningfully benefit practitioners and researchers working on domain adaptation of large language models.

We thank the reviewers and ACs once again for their constructive feedback and for helping us strengthen the paper. We hope these improvements address the concerns raised and further highlight the value of our work.

---

### Meta-Review · Area_Chair_47yQ · 2025-12-31

**Summary:**

The reviewers generally agree that the paper makes a solid and timely contribution by revisiting the widely held assumption that domain-specific supervised fine-tuning (SFT) inevitably degrades general capabilities of large language models. Across reviews, the core empirical finding, i.e., sufficiently small learning rates can largely mitigate catastrophic forgetting while retaining strong domain performance, is viewed as sound, clearly presented, and practically relevant. The compression- and information-theoretic analysis is consistently highlighted as a strength that helps explain observed phenomena, and the proposed Token-Adaptive Loss Reweighting (TALR) method is seen as a theoretically grounded extension that further improves the trade-off in higher learning-rate regimes.

At the same time, several reviewers express reservations about novelty and scope. In particular, concerns center on (i) whether the low-learning-rate insight is sufficiently novel given prior continual learning wisdom, (ii) whether TALR provides clear benefits beyond simple low-LR SFT in the most commonly used regimes, (iii) limited validation on larger-scale models (>8B) and architectures beyond decoder-only LLMs, and (iv) clarity and consistency of empirical advantages across models and settings. The rebuttal substantially strengthened the paper by adding new ablations, learning-rate sweeps, sequential-domain experiments, and clarifications of practical relevance, which alleviated many, but not all, of these concerns.

**Reviewer Concerns:**

Concerns Largely Addressed by the Rebuttal

- Effectiveness and robustness of TALR:
  The authors provided additional ablations, learning-dynamics analyses, and sequential domain experiments. These demonstrate that TALR is not ad hoc, that hard tokens are eventually learned, and that dynamic τ is superior to fixed alternatives (Reviewer T4Nt).

- Learning-rate sweep and over-training concerns:
  Expanded learning-rate sweeps and performance-vs-epoch plots clarified that the main conclusions are not artifacts of over-training and that the trade-off is driven primarily by LR magnitude (Reviewers 9brq, mWRP).

- Practical relevance of TALR despite small-LR SFT:
  By surveying SFT learning rates used in representative modern LLMs, the authors showed that many real systems use larger LRs where TALR offers clear benefits, strengthening the practical case (Reviewers mWRP, T4Nt).

- Sequential/continual domain adaptation:
  New experiments demonstrated that the approach preserves earlier domain performance under sequential SFT, directly addressing reviewer questions (Reviewer XUav).

Concerns Still Partially Outstanding

- Novelty relative to prior work:
  Some reviewers remain unconvinced that the low-learning-rate finding is sufficiently novel, viewing it as an expected heuristic from continual learning, even if its effectiveness in LLMs is better documented here (Reviewers mWRP, T4Nt).

- Scope and scalability:
  All experiments remain limited to decoder-only models up to 8B parameters. Generalization to larger dense models, MoE architectures, or encoder–decoder settings is still unverified and left as future work (Reviewers XUav, T4Nt).

- Consistency of gains across models:
  While clarified, some variability remains in how strongly TALR outperforms baselines across different base models, which may still leave questions about model-dependence (Reviewer 9brq).

**Reviewer Scores:**

- Reviewer XUav:
  Initial score: 6 → Updated estimate: 6 or 8
  (Explicitly indicated willingness to raise the score after rebuttal though this is around when the OpenReview incident happened; concerns largely addressed.)

- Reviewer 9brq:
  Initial score: 6 → Updated estimate: 6
  (Generally positive both before and after rebuttal; remaining concerns are relatively minor.)

- Reviewer mWRP:
  Initial score: 4 → Updated estimate: 6
  (Engaged deeply in discussion; rebuttal addressed novelty, LR regimes, and practical relevance sufficiently to justify moving above threshold.)

- Reviewer T4Nt:
  Initial score: 4 → Updated estimate: 4 or 6
  (Many technical concerns addressed through additional analyses and ablations, though novelty and scale remain reservations.)

---

### Decision · Program_Chairs · 2026-01-26

Accept (Poster)